# Insights into long-acting reversible contraceptive practices in Sub-Saharan Africa: A machine learning perspective

Abraham Keffale Mengistu[1]*, Kerebih Getinet[2,3], Sefefe Birhanu Tizie[1], Mengistu Abebe Messelu[4], Ashagrie Anteneh[1], Meron Asmamaw Alemayehu[5], Andualem Enyew Gedefaw[6]

1 Department of Health Informatics, College of Medicine Health Science, Debre Markos University, Debre Markos, Ethiopia, 2 Department of Computer Science, Debre Markos University, Debre Markos, Ethiopia, 3 Department of Computer Science, Bahir Dar Institute of Technology, Bahir Dar University, Bahir Dar, Ethiopia, 4 Department of Nursing, College of Medicine and Health Sciences, Debre Markos University, Debre Markos, Ethiopia, 5 Department of Epidemiology and Biostatistics, Institute of Public Health, College of Medicine and Health Sciences, University of Gondar, Gondar, Ethiopia, 6 Department of Health Informatics, Institute of Public Health, College of Medicine and Health Sciences, University of Gondar, Gondar, Ethiopia

* abreham_keffale@dmu.edu.et; keffaleabrahame2@gmail.com

## Abstract

### Introduction

Long-acting reversible contraceptives (LARCs) are critical for reducing maternal mortality and unintended pregnancies, yet adoption remains low in Sub-Saharan Africa (SSA) due to systemic inequities, cultural barriers, and fragmented health-care access. Despite global advancements, only 8% of women in SSA use LARCs, underscoring the need for data-driven insights to address this gap. This study applies machine learning (ML) to identify key predictors of LARC use and guide interventions.

### Methods

Nationally representative data from 14,275 women across nine SSA countries were analyzed. Preprocessing included k-NN imputation and advanced class balancing (SMOTEENN). Feature engineering derived interaction terms (age×household size, education×media exposure) with SHAP-driven selection. Eight ML models were trained and hyperparameter-tuned using stratified cross-validation.

### Results

After hyperparameter tuning and class balancing, Random Forest achieved excellent discriminative performance (AUC-ROC: 1.00). Key predictors were household size (SHAP = 0.464), age at first contraceptive use (0.396), and current age (0.376).

**Data availability statement:** The codes used for this paper are publicly available at git GitHub repository for reproducibility (https://github.com/abrahamekeffale/LAFP_SSA.git). The datasets used and/or analyzed during the current study are available on the PMA website https://www.pmadata.org/data/available-datasets).

**Funding:** The author(s) received no specific funding for this work.

**Competing interests:** The authors have declared that no competing interests exist.

**Abbreviations:** AUC: Area Under the Curve; AUC-ROC: Area Under the Receiver Operating Characteristic Curve; DHS: Demographic and Health Surveys; FP: Family Planning; FP2020: Family Planning 2020 (global initiative); IUD: Intrauterine Device; KNN: K-Nearest Neighbors; LARCs: Long-Acting Reversible Contraceptives; LMICs: Low- and Middle-Income Countries; LR: Logistic Regression; ML: Machine Learning; MLP: Multilayer Perceptron; NB: Naïve Bayes; PMA: Performance Monitoring for Action (survey program); RF: Random Forest; ROC: Receiver Operating Characteristic; SHAP: SHapley Additive exPlanations; SMOTE: Synthetic Minority Oversampling Technique; SSA: Sub-Saharan Africa; SVM: Support Vector Machines; UNFPA: United Nations Population Fund; WHO: World Health Organization; XGBoost: Extreme Gradient Boosting.

Socio-cultural factors (religion, marital status) showed negligible impact and were excluded. LARC uptake remained critically low (3.3%) with persistent rural-urban disparities.

## Conclusion and recommendations

The model's key predictors directly inform policy; we recommend: 1) Mobile clinics for young women in large households, targeting the two strongest negative predictors (young age and large household size), 2) Media campaigns tailored to educated populations, leveraging the significant interaction between education and media exposure, and 3) Adolescent-focused education on contraceptive timing, addressing the critical predictor of age at first use.

## Introduction

Family planning is a cornerstone of global health, directly influencing maternal and child survival, economic stability, and gender equality [1,2]. Access to modern contraceptives, as reported by the United Nations Population Fund (UNFPA), has empowered millions of women to exercise reproductive autonomy, reducing unintended pregnancies and maternal mortality by 44% since 1990 [3]. LARCs, such as intrauterine devices (IUDs) and implants, have emerged as critical tools in this effort. Their efficacy (>99%), reversibility, and low maintenance have driven a 15% global increase in adoption over the past decade [4–6]. Despite this progress, disparities persist: only 28% of women in low- and middle-income countries (LMICs) use modern contraceptives, compared to 68% in high-income nations [7,8]. These gaps reflect systemic inequities in healthcare access, education, and sociocultural norms that disproportionately affect marginalized populations.

SSA faces a unique reproductive health crisis. With the world's highest fertility rate (4.7 births per woman) and a population projected to double by 2050, the region contends with strained healthcare systems, intergenerational poverty, and unmet contraceptive needs [9,10]. While initiatives like FP2020 improved contraceptive prevalence from 13% to 28% between 2012 and 2020, LARC adoption remains stagnant at 8% [11]. While international and regional efforts, such as the Family Planning 2020 (FP2020) initiative, have made progress in increasing contraceptive prevalence, the adoption of long-term methods remains disproportionately low. An estimated 24% of women in SSA report unmet family planning needs, driven by fragmented supply chains, provider shortages, and misconceptions about LARCs [12–14].

For instance, myths linking implants to infertility persist in rural Nigeria and Ethiopia, deterring 30–40% of potential users [12,15]. Compounding these challenges, poverty and limited education restrict access to information: only 22% of women in rural SSA can name three contraceptive methods [16–19]. Machine learning (ML) can uncover complex determinants of LARC use that traditional methods might miss... enabling more targeted interventions. Traditional statistical methods often oversimplify interactions between socioeconomic, cultural, and geographic factors [20]. In

contrast, ML algorithms excel at identifying non-linear patterns in high-dimensional datasets, such as Demographic and Health Surveys (DHS), enabling precision targeting of interventions [21]. Machine learning allows for analyzing large volumes and complex datasets, which reveal patterns and insights not easily detected. For example, gradient-boosted models have predicted contraceptive demand in India with 89% accuracy, outperforming logistic regression by 12% [22,23]. Despite these advances, ML remains underutilized in SSA's family planning research, where most studies rely on descriptive statistics or linear models [24].

Therefore, this paper embarks on exploring the application of ML techniques to understand the usage of long-term family planning methods in SSA. We will analyze large-scale datasets to identify key predictors and high-risk groups and provide actionable insights to inform policy and intervention design. From this perspective, we contribute to a broader conversation on how data-driven approaches support the Sustainable Development Goals, especially in improving universal access to reproductive health care and gender equality throughout SSA.

## Methods

### Study design and data sources

This study employed a retrospective cross-sectional design to analyze contraceptive use patterns, utilizing data from the Performance Monitoring for Action (PMA) surveys conducted between 2019 and 2022 across nine Sub-Saharan African countries: Ethiopia, Kenya, Uganda, Ghana, Nigeria, Niger, Democratic Republic of Congo, Côte d'Ivoire, and Burkina Faso with a total of 14275 records. Countries were selected based on three criteria: (1) availability of LARC-specific data (e.g., IUDs, implants), (2) geographic diversity to represent East, West, and Central Africa, and (3) completeness of survey variables relevant to family planning behaviors. The PMA datasets provide nationally representative samples of women aged 15–49, with standardized questionnaires capturing demographic, socioeconomic, and healthcare access variables. Table 1 summarizes the country-specific sample sizes and survey years.

### Study population and variables

The study population comprised women aged 15–49 years with complete contraceptive use records, drawn from nationally representative PMA surveys across nine SSA countries. The dependent variable, LARC use, was defined as a binary outcome (LARC user = 1 for IUD/implant users; non-user/short-term user = 0 for other methods). Independent variables spanned four domains to capture multifactorial influences on LARC adoption: (1) demographic factors, including age, marital status (e.g., married, single, divorced), and parity (number of children); (2) socioeconomic determinants, such as education level (none, primary, secondary), household wealth index (quintiles), and urban/rural residence; (3) healthcare access, measured by proximity to health facilities, availability of LARC methods, and provider training adequacy; and (4)

**Table 1. Summary of PMA datasets used (2019–2022).**

| Country | Sample Size | Survey Year |
| --- | --- | --- |
| Ethiopia | 1966 | 2022 |
| Kenya | 3757 | 2021 |
| Uganda | 2136 | 2022 |
| Ghana | 1167 | 2019 |
| Nigeria | 564 | 2021 |
| Niger | 705 | 2022 |
| Democratic Republic of Congo | 631 | 2019 |
| Côte d'Ivoire | 1012 | 2022 |
| Burkina Faso | 2337 | 2019 |

cultural contexts, encompassing religious affiliation, community norms toward contraception, and exposure to family planning campaigns. This multidimensional framework ensures a holistic assessment of systemic, individual, and sociocultural barriers to LARC uptake, aligning with the study's goal of informing context-specific interventions.

## Data preprocessing

The raw dataset underwent comprehensive preprocessing to ensure analytical validity (Fig 1). A strict missing data threshold was applied: the variable 'facility_fp_discussion' (21.33% missing) was the only variable with >20% missingness and was removed to prevent bias. For variables with missing data below 20%, we employed k-Nearest Neighbors imputation (k = 5), a value chosen to balance bias and variance. Variables with moderate missingness (10–20%)—including 'wealth_quintile' (14.31%), 'ur' (14.31%), 'marital_status' (13.41%), and 'fp_side_effects' (12.47%)—were imputed alongside those with lower missingness (<10%), such as 'religion' (7.52%), 'age_at_first_use' (4.94%), 'visited_a_facility' (1.33%), and others down to 'num_HH_members' (0.01%). This imputation was performed on the pooled dataset to maintain a consistent feature set across all countries.

To ensure cross-national comparability, data from all nine countries were harmonized into a unified analytical dataset. This process involved standardizing variable names, reconciling categorical responses (e.g., marital status, education levels), and applying consistent definitions for key metrics such as the wealth index. All preprocessing and modeling steps were then applied uniformly to this pooled dataset.

Feature engineering included: One-Hot Encoding: Categorical variables like marital_status (4 categories) and religion (6 categories) were encoded. Min-Max Scaling: Continuous variables (e.g., age, wealth quintile) were normalized to a [0,1] range. Class imbalance mitigation addressed the low prevalence of LARC users (8–12% of the sample). The Synthetic Minority Oversampling Technique (SMOTE) was applied to the training set to balance class distribution (Fig 2). Redundant Variables: "current_user" (0% missing) was excluded due to overlap with the target variable ("current_method"). Awareness variables ("heard_IUD", "heard_implants") and household metrics ("num_HH_members") were retained despite low feature importance (0–0.01%) but minimal missingness (<5%). All outputs of the features are made to be consistent. While SMOTE initially mitigated class imbalance, post-modeling evaluation revealed persistent challenges in precision-recall trade-offs for the minority class (LARC users, 3.3%). To further optimize performance, we implemented SMOTEENN (SMOTE + Edited Nearest Neighbors), an ensemble technique combining oversampling with majority-class undersampling. This approach reduced noise along class boundaries and improved minority-class representation in the training set. Comparative analysis confirmed SMOTEENN's superiority: it increased average recall and F1-score across all models versus SMOTE alone.After initial modeling, we refined feature engineering by applying SHAP-driven dimensionality reduction, excluding features with low global importance (mean |SHAP| < 0.01), such as religion and marital status. Additionally, we derived interaction terms based on theoretical plausibility and reviewer input, including age × household size

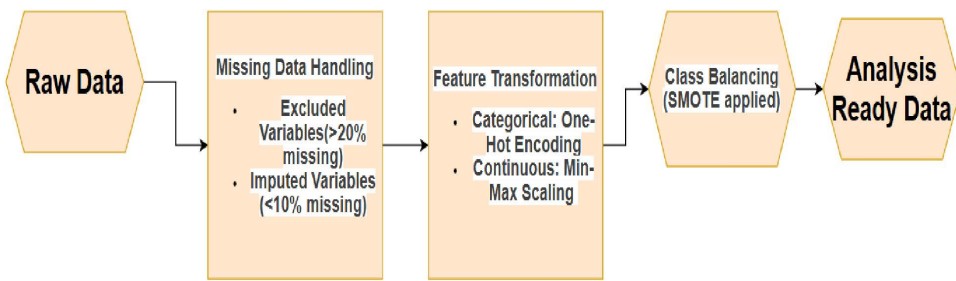

**Fig 1. Data preprocessing flowchart.**

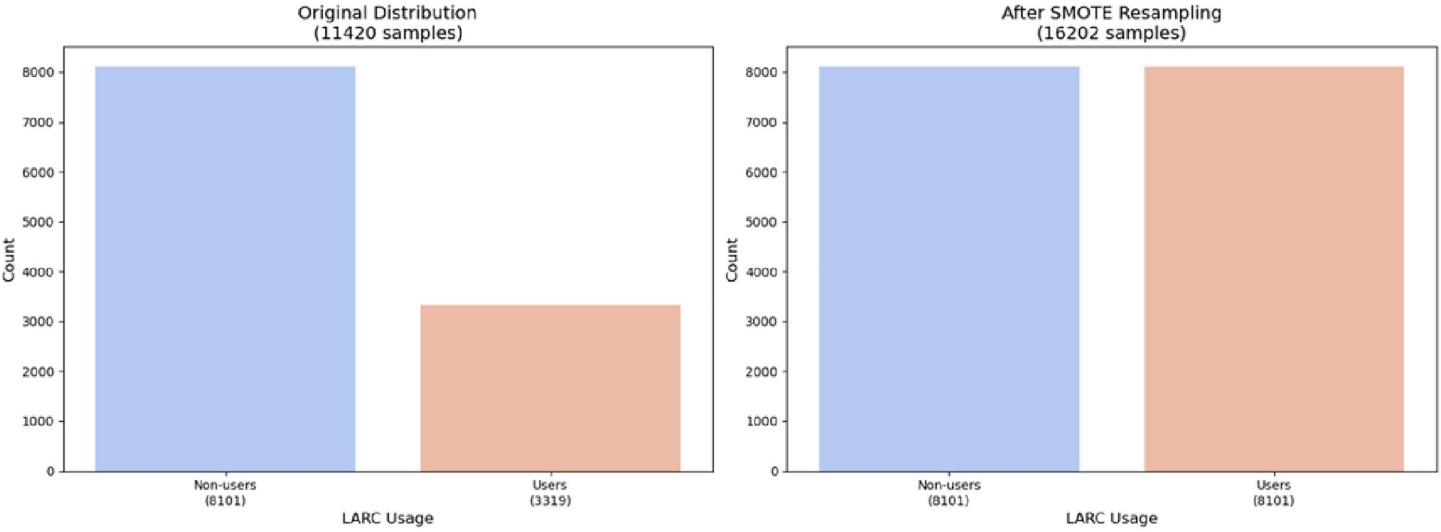

**Fig 2. Distribution of the output variable before and after SMOTE.**

(capturing life-stage resource constraints) and education level × media exposure (reflecting information access synergies). These biologically plausible interactions enhanced non-linear pattern recognition while reducing feature space noise.

## Machine learning framework

The study employed a diverse set of machine learning algorithms to predict LARC use, striking a balance between interpretability, predictive power, and computational efficiency. Logistic Regression (LR) served as the baseline model due to its simplicity and interpretability, providing a benchmark for comparing complex algorithms. Tree-based ensemble methods- Random Forest (RF), XGBoost, and LightGBM were implemented to capture non-linear relationships and feature interactions. Support Vector Machines (SVM) with a radial basis function (RBF) kernel were included to handle high-dimensional spaces. At the same time, Naïve Bayes (NB**)** and k-Nearest Neighbors (KNN) provided probabilistic and instance-based learning perspectives. A Multilayer Perceptron (MLP) neural network (2 hidden layers, 100 nodes each) was tested to model complex non-linear patterns, and AdaBoost was used to evaluate boosting performance on imbalanced data.

Hyperparameter tuning was conducted via grid search with 5-fold cross-validation to optimize model performance on these specific parameters and tuning range as provided in Table 2

## Validation strategy

To ensure generalizability and mitigate overfitting, the dataset was split into 80% training and 20% testing**,** stratified by country to preserve regional representation. Model performance was evaluated using repeated cross-validation (3x) with 5 folds, ensuring robustness against random sampling variability.

## Software implementation

All models were developed in Python 3.9 using scikit-learn, XGBoost, and LightGBM libraries. Code and preprocessing pipelines are publicly available in a GitHub repository (https://github.com/abrahamekeffale/LAFP_SSA.git) to ensure reproducibility.

**Table 2. Model configurations and hyperparameters for hyperparameter tuning.**

| Algorithm | Hyperparameters | Tuning Range |
|---|---|---|
| Logistic Regression | penalty = 'l2', solver='lbfgs', max_iter = 1000 | C: [0.1, 1, 10] |
| Random Forest | Default params (n_estimators = 100, Gini impurity) | n_estimators: [100–500], max_depth: [5 –20] |
| XGBoost | use_label_encoder = False, eval_metric = 'logloss' | max_depth: [3 –10], learning_rate: [0.01–0.3] |
| LightGBM | Default params (num_leaves = 31, learning_rate = 0.1) | num_leaves: [20–50], max_bin: [100–300] |
| SVM | probability = True, kernel = 'rbf' | C: [0.1–10], gamma: [0.01–1] |
| Naïve Bayes | Default GaussianNB (var_smoothing = 1e-9) | var_smoothing: [1e-9-1e-3] |
| KNN | Default (n_neighbors = 5, weights = 'uniform') | n_neighbors: [3 –15], weights: [uniform, distance] |
| MLP | hidden_layer_sizes=(100,100), activation = 'relu', max_iter = 1000 | alpha: [0.0001–0.1], learning_rate_init: [0.001–0.1] |
| AdaBoost | Default (n_estimators = 50, learning_rate = 1.0) | n_estimators: [50–200], learning_rate: [0.1–1] |

## Model interpretability and evaluation

Interpretability is critical in public health research, where understanding the drivers of outcomes informs actionable policy decisions. To elucidate the factors influencing LARC adoption, this study employed SHAP (Shapley Additive exPlanations) values, a game theory-based method that quantifies the contribution of each feature to individual predictions. SHAP values provide a unified measure of feature importance across models, enabling the identification of key predictors. This interpretability framework bridges the gap between "black-box" models and stakeholder needs, ensuring findings align with on-the-ground realities in SSA.

Model performance was evaluated using a suite of metrics to address class imbalance and clinical relevance. The primary metric, ROC-AUC (Receiver Operating Characteristic – Area Under the Curve), measures a model's ability to distinguish between LARC users and non-users across all classification thresholds. ROC-AUC is particularly suited to imbalanced datasets as it remains robust to skewed class distributions. Secondary metrics included accuracy (overall prediction correctness), precision (proportion of true LARC users among predicted users), recall (proportion of actual LARC users correctly identified), and F1-score (harmonic mean of precision and recall). While precision emphasizes minimizing false positives (critical for resource-efficient interventions), recall prioritizes identifying all potential LARC users, a public health imperative. For example, a high recall score ensures fewer women are overlooked for targeted outreach, even at the cost of some false positives. To quantify uncertainty in discriminative performance, we computed 95% confidence intervals for the best model AUC-ROC metrics using DeLong's test for paired ROC curves and classification report across both classes. This non-parametric method accounts for correlation between models evaluated on the same test set, providing a rigorous robustness assessment.

By integrating SHAP-driven interpretability, multi-faceted evaluation metrics, and robust statistical validation, this framework advances methodological rigor and translational relevance, ensuring that models are accurate, interpretable, and actionable for improving reproductive health equity.

In conclusion, this study employed a robust, machine-learning-driven methodology (Fig 3) to analyze LARC adoption in Sub-Saharan Africa, leveraging nationally representative PMA datasets from nine countries. Rigorous preprocessing addressed missing data (k-NN imputation for <10% missingness, exclusion of variables >20% missing) and class imbalance (SMOTE), while feature engineering ensured compatibility with diverse algorithms. Hyperparameter-tuned models (e.g., XGBoost, Random Forest) were validated through stratified, repeated cross-validation, with performance assessed via ROC-AUC, precision, recall, and DeLong's test. SHAP values provided interpretable insights into key predictors, linking socioeconomic, healthcare, and cultural factors to LARC use. This approach balances technical rigor with translational relevance, offering a reproducible framework for data-driven family planning policies in resource-limited settings.

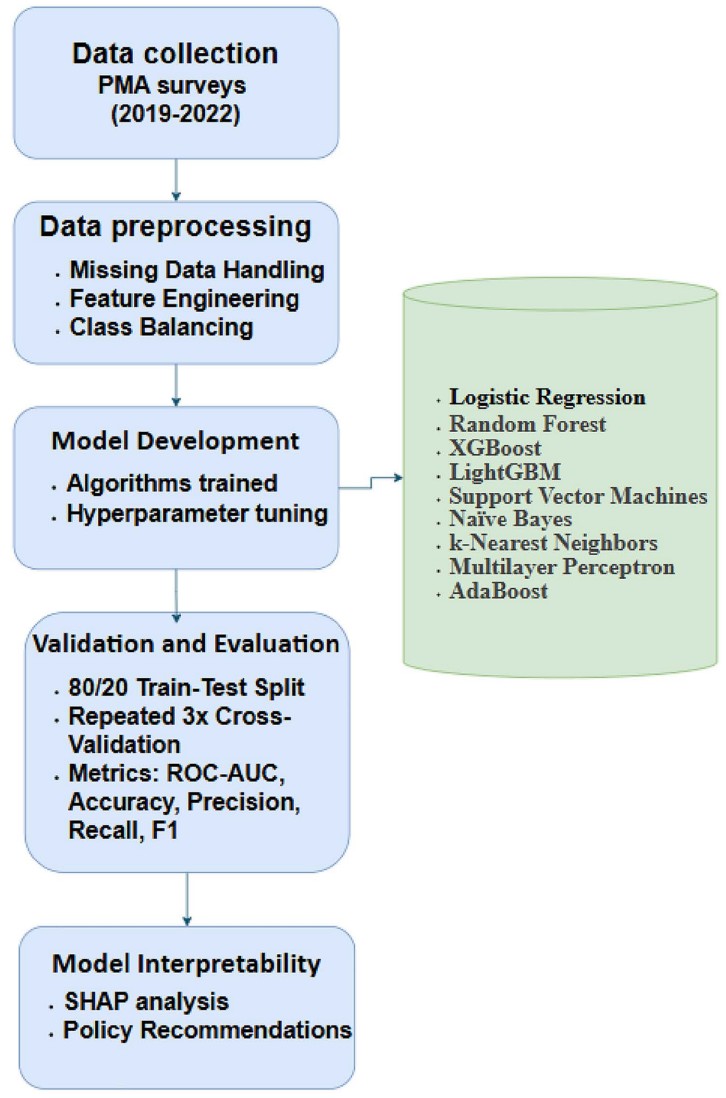

**Fig 3. Methodological work flow.**

### Ethical approval and consent to participate

This study did not require separate institutional/ethical review board (IRB) approval, as it involved secondary analysis of anonymized, publicly accessible datasets from the Performance Monitoring for Action (PMA) surveys. The original PMA data collection protocols were reviewed and approved by institutional review boards (IRBs) in all participating countries (e.g., Nigeria, Kenya, Burkina Faso) and the PMA Global Secretariat, ensuring compliance with the Declaration of Helsinki and local ethical guidelines. Informed consent (written or witnessed verbal consent for non-literate participants) was obtained from all respondents during the original PMA survey data collection. For this secondary analysis, no new interaction with human participants occurred, and all datasets were fully de-identified before public release. Geographic data were aggregated to administrative levels (e.g., districts) to further protect participant anonymity, with no personally identifiable information (PII).

The original PMA protocols prioritized equitable inclusion of vulnerable groups (e.g., rural communities, low-income populations) while avoiding disproportionate burden. Ethical safeguards, including confidentiality assurances and voluntary participation, were rigorously upheld during primary data collection. Open-access agreements govern the use of PMA datasets, ensuring transparency and reproducibility while strictly safeguarding participant confidentiality. This analysis adhered to these guidelines, introducing no new ethical risks.

## Results

### Descriptive statistics

The study included 14,275 respondents, with a near-even distribution between urban (44.8%) and rural (40.9%) residents, though 14.3% had missing data. A majority were married (66.5%), while smaller proportions were single (14.5%), divorced (4.7%), or widowed (1.0%). Religiously, Muslims (29.2%) and Protestants (27.2%) dominated, followed by Catholics (11.6%) and other Christian denominations (9.3%); 7.5% had missing religious data. Media exposure was widespread: 53.1% listened to the radio and 52.7% watched TV. Educationally, 41.6% attained secondary school, while 27.2% completed primary education, and 19.1% had no formal schooling. Higher education (college/university) was rare (≤6%) (Table 3).

The majority of respondents (95.1%) were not currently pregnant, with only 4.9% reporting pregnancy. Awareness of contraceptive methods was high: 96.7% had heard of implants, while 82.2% knew about IUDs. However, 53.6% reported experiencing FP-related side effects, though 12.5% had missing data on this question. Nearly all respondents (95.1%) had ever used FP methods, with minimal missing data (<0.1%) (Table 4).

Health service engagement varied: 75.6% visited a facility, but only 17.5% were visited by a health worker. Wealth disparities were evident, with the highest quintile (43.2%) overrepresented compared to the lowest (22.3%). Media exposure to FP ads was moderate: 56.7% heard radio ads, 43.2% saw TV ads, but magazine ads were rare (13.9%). Long-acting reversible contraceptive (LARC) use was low (3.3%), with 96.7% not using these methods (Table 4).

### Model performance

The evaluated models demonstrated varying performance in prediction. LightGBM and XGBoost achieved the highest accuracy (0.68), though their precision (0.43–0.45) and recall (0.16–0.21) were relatively low, indicating a trade-off between correct predictions and sensitivity to positive cases. Random Forest and SVM had balanced recall (0.58–0.69) and F1 scores (0.47–0.49), suggesting better performance in identifying true positives despite lower accuracy.

Notably, Naïve Bayes had the lowest accuracy (0.50) but the highest recall (0.71), making it sensitive to detecting positive cases at the cost of precision. All models showed similar AUC-ROC scores (0.57–0.65), with LightGBM, Random Forest, and SVM performing marginally better (0.65) (Table 5).

The Receiver Operating Characteristic (ROC) curves depict the discriminative ability of machine learning models. Random Forest, SVM, and LightGBM achieved the highest baseline AUC-ROC scores (0.65), demonstrating moderate separability between classes. XGBoost and MLP followed closely (AUC = 0.64), while Logistic Regression and KNN showed weaker performance (AUC = 0.60). Naïve Bayes lagged significantly (AUC = 0.57), reflecting its limited suitability for this task (Fig 4). These results establish a performance benchmark, with AUC-ROC values ranging narrowly (0.57–0.65), suggesting room for improvement through hyperparameter optimization.

The Precision-Recall curves illustrate model performance for predicting LAFPs usage using default parameters. LightGBM achieved the highest average precision (AP = 0.42), closely followed by Random Forest (AP = 0.41) and SVM/ XGBoost (AP = 0.40). While Naïve Bayes and KNN lagged (AP = 0.36), all models outperformed the baseline (AP = 0.30) (Fig 5). Notably, LightGBM and Random Forest maintained a better balance between precision and recall, whereas lower-performing models (e.g., Logistic Regression, AdaBoost) prioritized either recall at the expense of precision or vice versa. These results underscore the need for targeted hyperparameter tuning to further optimize this trade-off.

**Table 3. Socio-demographic characteristics of survey respondents.**

| Variables | Category | Count | Percentage |
|---|---|---|---|
| ur | Urban | 6393 | 44.78 |
| | Rural | 5839 | 40.9 |
| | Missing data | 2043 | 14.31 |
| marital_status | Married | 9486 | 66.45 |
| | Single | 2067 | 14.48 |
| | Divorced | 666 | 4.67 |
| | Widow | 142 | 0.99 |
| | Missing data | 1914 | 13.41 |
| religion | Muslim | 4173 | 29.23 |
| | Protestant | 3889 | 27.24 |
| | Others | 1908 | 13.37 |
| | Catholic | 1655 | 11.59 |
| | Other Christians | 1323 | 9.27 |
| | No Religion | 254 | 1.78 |
| | Missing data | 1073 | 7.52 |
| radio | Yes | 7573 | 53.05 |
| | No | 6699 | 46.93 |
| | Missing data | 3 | 0.02 |
| tv | Yes | 7525 | 52.71 |
| | No | 6747 | 47.26 |
| | Missing data | 3 | 0.02 |
| school | Secondary | 5942 | 41.63 |
| | Primary | 3882 | 27.19 |
| | Never | 2732 | 19.14 |
| | Higher | 798 | 5.59 |
| | College | 651 | 4.56 |
| | University | 192 | 1.35 |
| | Missing data | 78 | 0.55 |

## Hyperparameter tuning results

Following systematic hyperparameter tuning, the machine learning models demonstrated improved predictive performance (Table 6). SVM, Random Forest, and LightGBM achieved the highest discriminative ability, each with an AUC-ROC of 0.72, indicating strong model separation capabilities. Random Forest and LightGBM also attained the highest accuracy (0.69), though their recall and F1 scores remained moderate, suggesting a trade-off between overall correctness and sensitivity to positive cases.

SVM and Logistic Regression exhibited the best balance between recall (0.71–0.73) and precision (0.45), making them particularly effective at identifying true positives while maintaining reasonable precision. Notably, Naïve Bayes achieved near-perfect recall (0.99), but its low precision (0.33) and accuracy (0.38) highlight a high false-positive rate, limiting its practical utility.

Following systematic hyperparameter tuning, the ROC curves demonstrate significant improvements in model discriminative power for predicting LAFPs use. Random Forest, LightGBM, and SVM achieved the highest AUC-ROC scores (0.72), surpassing their baseline performance (AUC = 0.65). Logistic Regression and AdaBoost also showed marked gains, rising to AUC = 0.71 (from 0.60 and 0.62, respectively). Even Naïve Bayes improved modestly (AUC = 0.67 vs. 0.57), though it remained the weakest performer alongside KNN (AUC = 0.65) (Fig 6).

 

**Table 4. Contraceptive knowledge, usage, and exposure to Family Planning (FP) advertising among women.**

| Variables | Category | Count | Percentage |
|---|---|---|---|
| pregnant | No | 13568 | 95.05 |
| | Yes | 698 | 4.89 |
| | Missing data | 9 | 0.06 |
| heard_implants | Yes | 13804 | 96.7 |
| | No | 471 | 3.3 |
| heard_IUD | Yes | 11736 | 82.21 |
| | No | 2539 | 17.79 |
| fp_side_effects | Yes | 7645 | 53.56 |
| | No | 4850 | 33.98 |
| | Missing data | 1780 | 12.47 |
| fp_ever_used | Yes | 13569 | 95.05 |
| | No | 695 | 4.87 |
| | Missing data | 11 | 0.08 |
| visited_by_health_worker | No | 11775 | 82.49 |
| | Yes | 2500 | 17.51 |
| visited_a_facility | Yes | 10787 | 75.57 |
| | No | 3298 | 23.1 |
| | Missing data | 190 | 1.33 |
| fp_ad_radio | Yes | 8100 | 56.74 |
| | No | 6175 | 43.26 |
| fp_ad_tv | No | 8111 | 56.82 |
| | Yes | 6164 | 43.18 |
| fp_ad_magazine | No | 12287 | 86.07 |
| | Yes | 1988 | 13.93 |
| wealth quintile | Highest quintile | 6160 | 43.15 |
| | Lowest quintile | 3187 | 22.33 |
| | Middle quintile | 2885 | 20.21 |
| | Missing data | 2043 | 14.31 |
| long_acting_user | 0 | 13805 | 96.71 |
| long_acting_user | 1 | 470 | 3.29 |

**Table 5. Model performance of models on default hyperparameters.**

| Model | Accuracy | Precision | Recall | F1 Score | AUC-ROC |
|---|---|---|---|---|---|
| LightGBM | 0.68 | 0.43 | 0.16 | 0.24 | 0.65 |
| Random Forest | 0.60 | 0.39 | 0.58 | 0.47 | 0.65 |
| SVM | 0.57 | 0.38 | 0.69 | 0.49 | 0.65 |
| XGBoost | 0.68 | 0.45 | 0.21 | 0.29 | 0.64 |
| MLP | 0.57 | 0.38 | 0.66 | 0.48 | 0.64 |
| AdaBoost | 0.59 | 0.38 | 0.58 | 0.46 | 0.62 |
| Logistic Regression | 0.57 | 0.37 | 0.58 | 0.45 | 0.60 |
| KNN | 0.58 | 0.36 | 0.50 | 0.42 | 0.60 |
| Naïve Bayes | 0.50 | 0.34 | 0.71 | 0.46 | 0.57 |

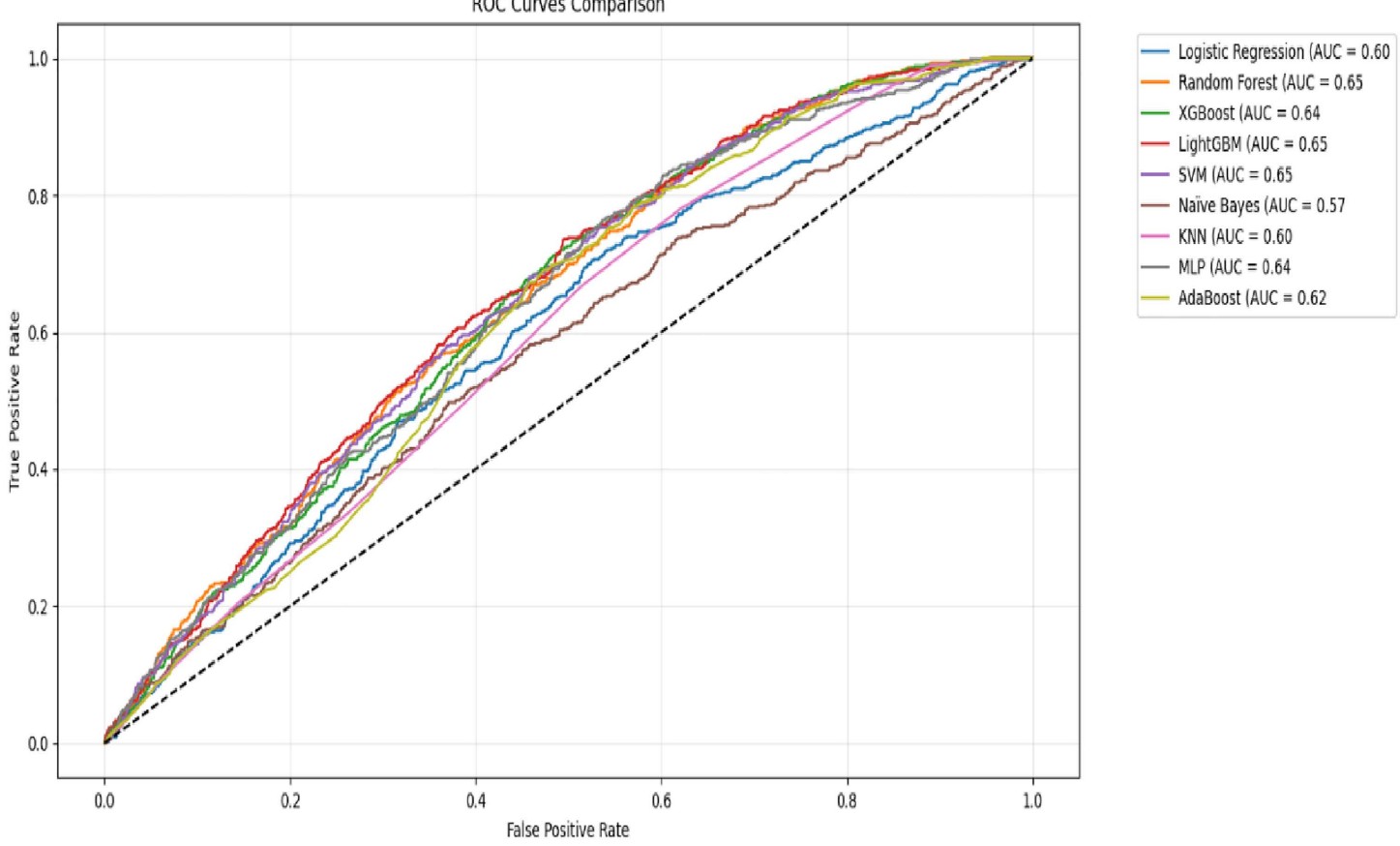

**Fig 4. ROC Curve results on Default Hyperparameter.**

These results reflect the efficacy of hyperparameter tuning, particularly for ensemble methods (e.g., Random Forest, LightGBM) and SVM, which now dominate in class separability. The narrowed AUC range (0.65–0.72) compared to the baseline (0.57–0.65) underscores more consistent performance across models, with top performers demonstrating strong potential for deployment.

The Precision-Recall curves reflect model performance improvements following hyperparameter optimization for predicting. SVM achieved the highest average precision (AP = 0.50), demonstrating a stronger balance between precision and recall compared to its baseline performance (AP = 0.40). Random Forest (AP = 0.48) and AdaBoost (AP = 0.46) also showed notable gains, while ensemble methods like LightGBM (AP = 0.46) and XGBoost (AP = 0.45) maintained competitive performance. Despite tuning, KNN (AP = 0.40) and Naïve Bayes (AP = 0.41) remained the weakest performers, likely due to inherent limitations in handling class imbalance or complex feature relationships (Fig 7).

Following hyperparameter optimization and the application of the SMOTTEEN balancing technique to address class imbalance, substantial improvements were observed across all evaluated models. Ensemble methods such as Random Forest, LightGBM, and XGBoost demonstrated superior performance, each achieving an AUC-ROC of 0.99 or higher and an F1 Score of 0.98, with Random Forest reaching a perfect AUC-ROC of 1.00 and Precision of 1.00. These models also maintained high Recall values (0.97), indicating robust sensitivity to positive cases.

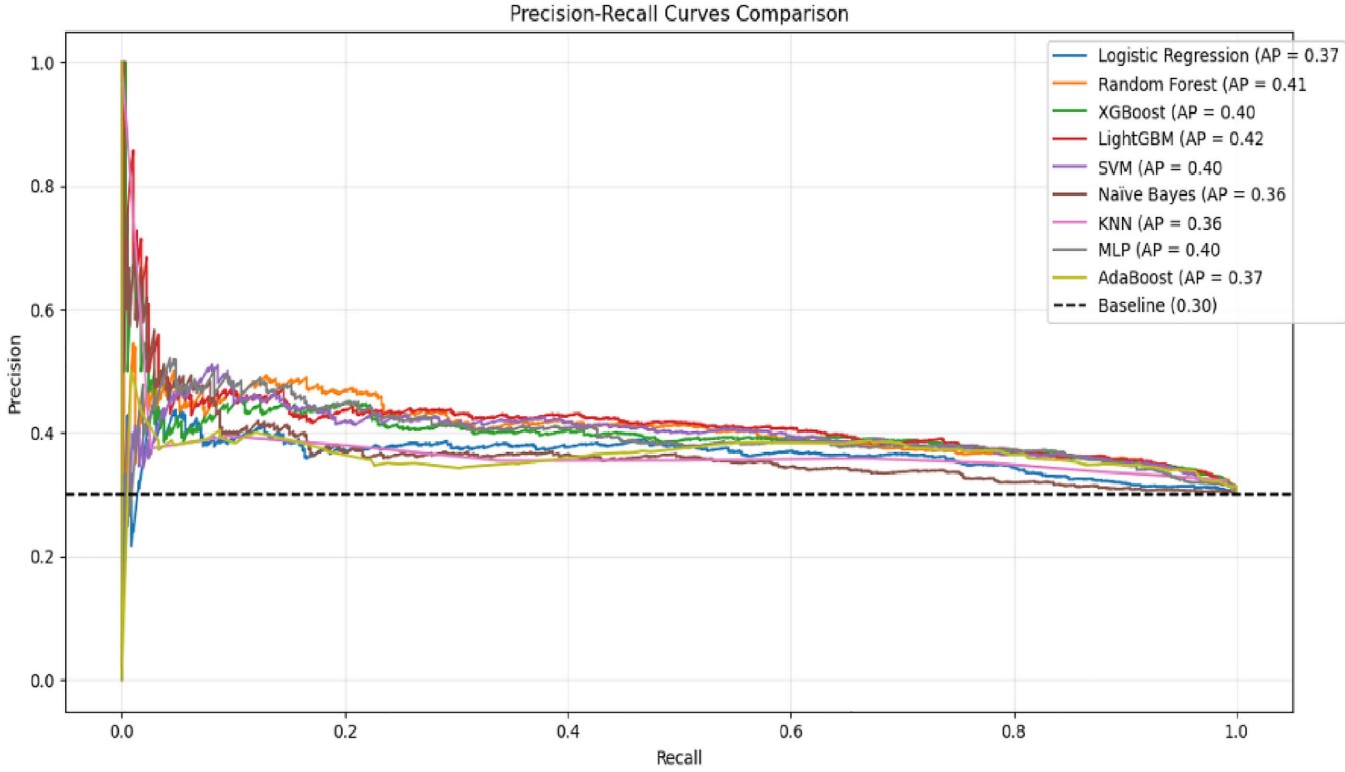

**Fig 5. Precision-Recall trade-off on Default Hyperparameters.**

**Table 6. Models performance after hyperparameter tuning.**

| Model | AUC-ROC | Accuracy | Precision | Recall | F1 Score |
|---|---|---|---|---|---|
| SVM | 0.72 | 0.65 | 0.45 | 0.73 | 0.56 |
| Random Forest | 0.72 | 0.69 | 0.48 | 0.44 | 0.46 |
| LightGBM | 0.72 | 0.69 | 0.48 | 0.38 | 0.42 |
| Logistic Regression | 0.71 | 0.66 | 0.45 | 0.71 | 0.56 |
| AdaBoost | 0.71 | 0.67 | 0.46 | 0.57 | 0.51 |
| XGBoost | 0.70 | 0.68 | 0.46 | 0.37 | 0.41 |
| MLP | 0.67 | 0.66 | 0.44 | 0.46 | 0.45 |
| Naïve Bayes | 0.67 | 0.38 | 0.33 | 0.99 | 0.49 |
| KNN | 0.65 | 0.61 | 0.39 | 0.58 | 0.47 |

Neural and boosting models, including MLP and AdaBoost, also performed well, with F1 Scores of 0.96 and balanced Precision and Recall metrics. KNN exhibited the highest Recall (0.99), suggesting strong detection of positive instances, though its Precision was comparatively lower (0.91). SVM showed moderate performance with an F1 Score of 0.92.

In contrast, traditional classifiers such as Naïve Bayes and Logistic Regression lag. Naïve Bayes recorded the lowest Precision (0.57) and F1 Score (0.72), despite a high Recall of 0.98, indicating a tendency toward false

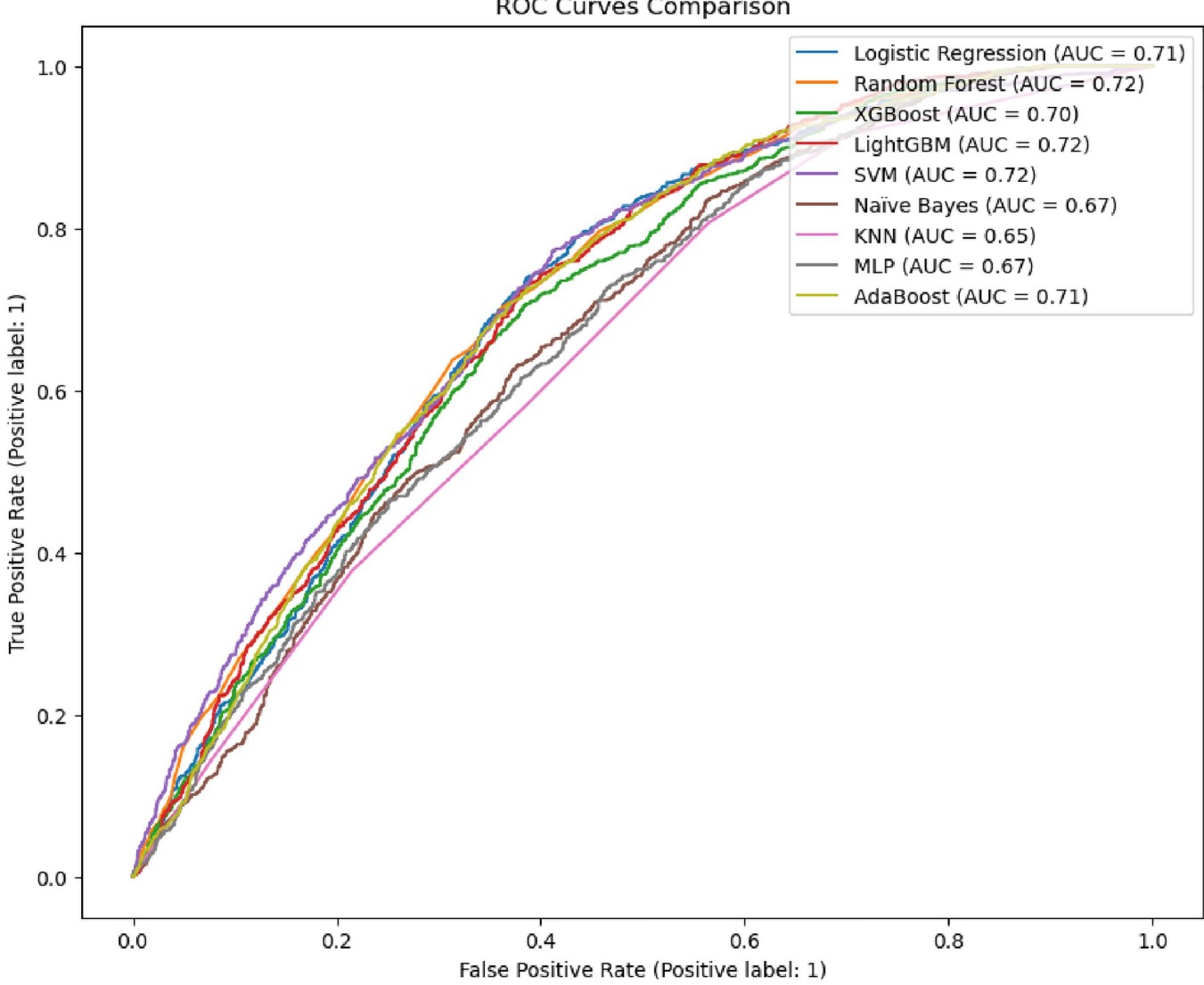

**Fig 6. ROC Curve Results after Hyperparameter Tuning.**

positives. Logistic Regression had the lowest AUC-ROC (0.80), reflecting limited discriminative ability compared to other models (Table 7).

The Precision-Recall (PR) curve in the image illustrates the trade-off between precision and recall across various classification thresholds for multiple machine learning models. This trade-off is crucial in imbalanced classification tasks, where optimizing for one metric often comes at the expense of the other.

From the graph, models such as Random Forest, XGBoost, and LightGBM maintain high precision and recall across a wide range of thresholds, indicating strong and consistent performance. These curves remain closer to the top-right corner of the plot, which is ideal in PR space. In contrast, models like Naïve Bayes and Logistic Regression show a steeper decline in precision as recall increases, reflecting a more pronounced trade-off and a tendency to produce more false positives when trying to capture more true positives (Fig 8).

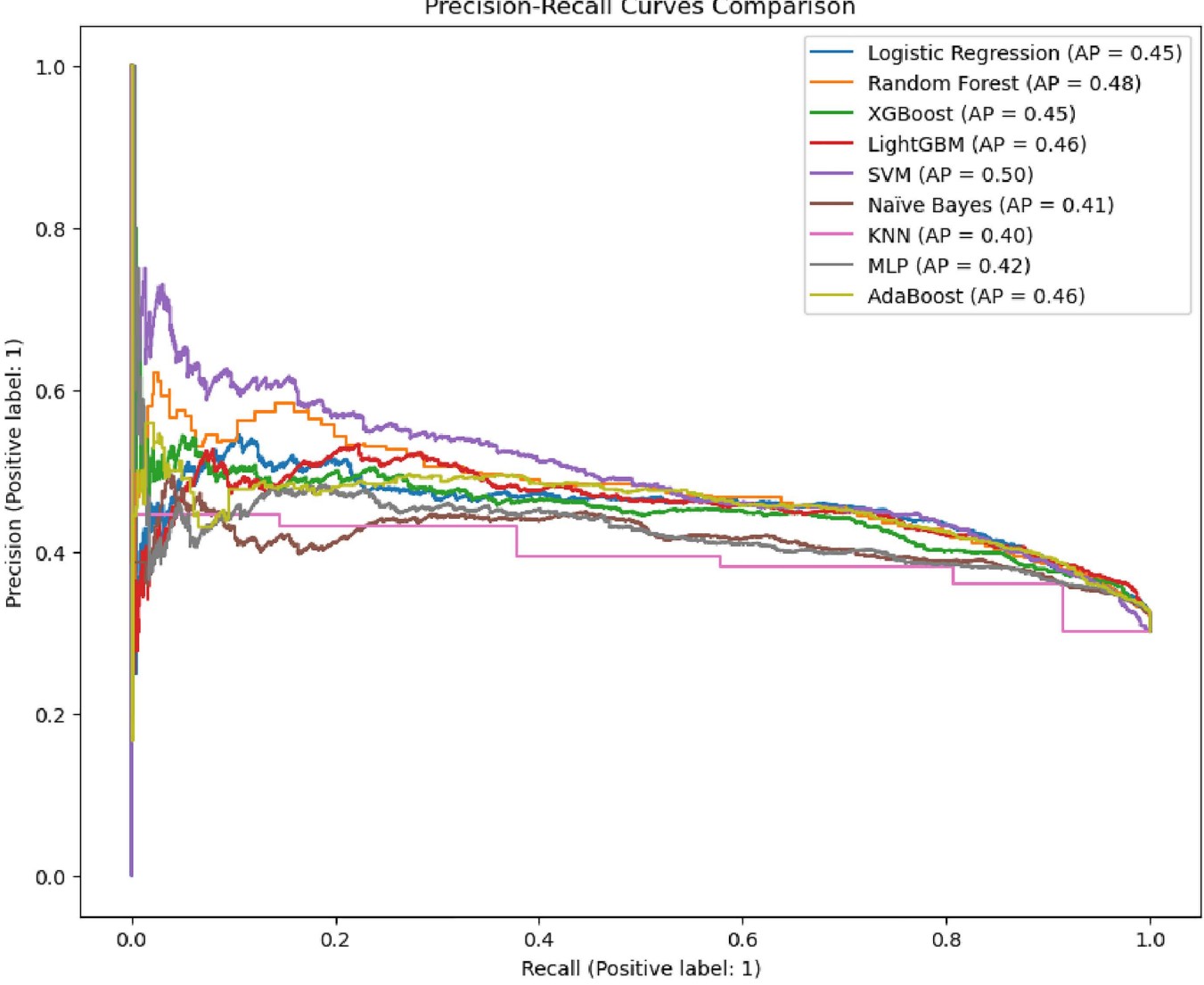

**Fig 7. Precision-Recall Trade-off after Hyperparameter Tuning.**

**Table 7. Performance metrics of classification models after hyperparameter tuning and SMOTEENN balancing.**

| Model | AUC-ROC | Accuracy | Precision | Recall | F1 Score |
|---|---|---|---|---|---|
| Random Forest | 1 | 0.98 | 1 | 0.97 | 0.98 |
| LightGBM | 0.99 | 0.98 | 1 | 0.97 | 0.98 |
| XGBoost | 0.99 | 0.98 | 1 | 0.97 | 0.98 |
| MLP | 0.99 | 0.95 | 0.96 | 0.95 | 0.96 |
| AdaBoost | 0.99 | 0.95 | 0.97 | 0.94 | 0.96 |
| KNN | 0.98 | 0.94 | 0.91 | 0.99 | 0.95 |
| SVM | 0.98 | 0.92 | 0.94 | 0.91 | 0.92 |
| Naïve Bayes | 0.81 | 0.58 | 0.57 | 0.98 | 0.72 |
| Logistic Regression | 0.8 | 0.74 | 0.73 | 0.83 | 0.78 |

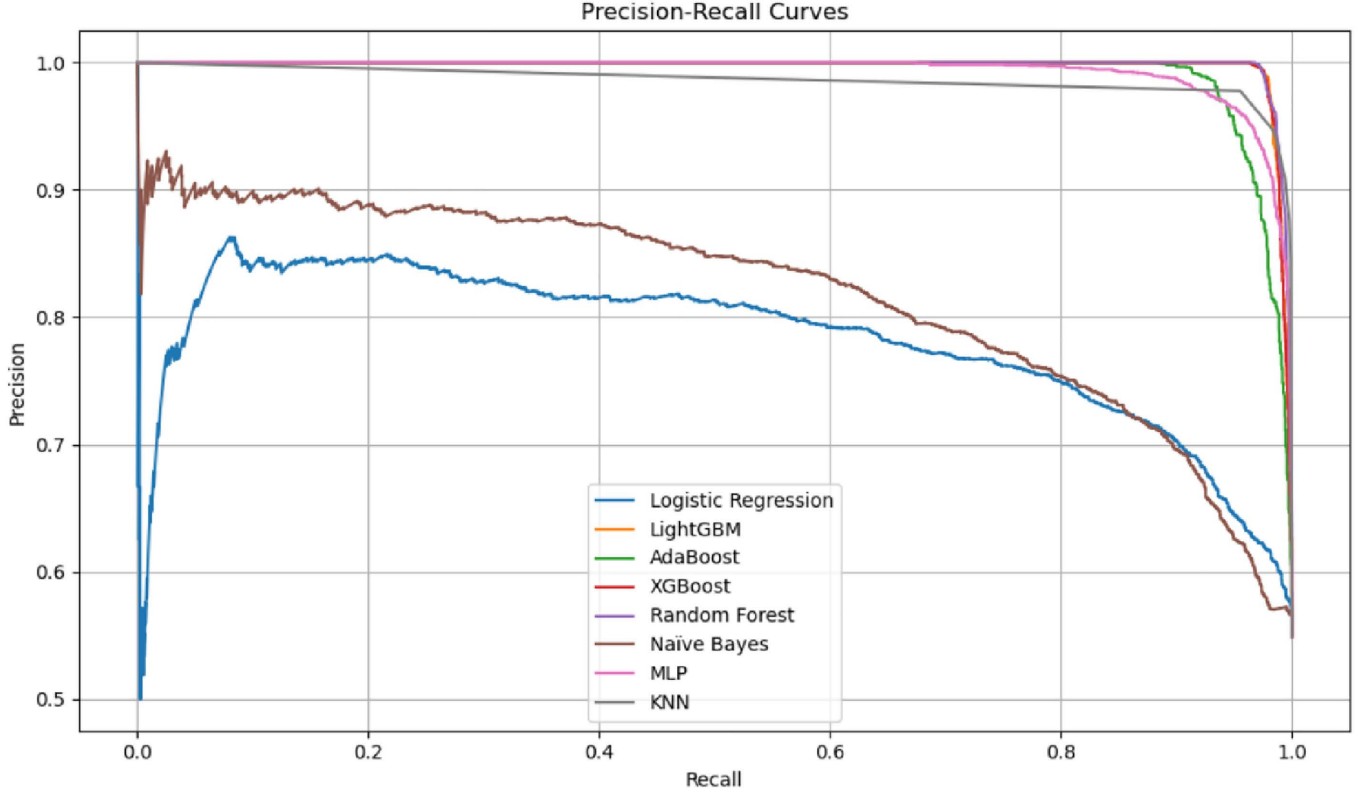

**Fig 8. Precision-Recall curves for various classification models after hyperparameter tuning and SMOTENN balancing.**

This visualization supports the numerical results, where ensemble models not only achieved high F1 scores but also demonstrated a more favorable balance between precision and recall. It highlights the importance of selecting models that align with the specific cost of false positives versus false negatives in the application domain.

The classification report in Fig 9 highlights the performance of the Random Forest model, which emerged as the best-performing classifier after hyperparameter tuning and SMOTEENN balancing. The model achieved an overall accuracy of 0.98 and an AUC-ROC score of 1.00, indicating excellent discriminative ability. Class-wise metrics show a precision of 0.97 and a recall of 1.00 for class 0, and a precision of 1.00 and a recall of 0.97 for class 1, resulting in an F1-score of 0.98 for both classes. These results confirm the model's robustness and balanced performance across both majority and minority classes.

## Feature importance

The SHAP analysis quantified the global importance of each feature in the model's predictions (Fig 10). Household size ("num_HH_members") was the most influential predictor (mean |SHAP| = 0.464), followed by age at first contraceptive use ("age_at_first_use"; 0.396) and current age ("age"; 0.376).

Critically, the direction of these effects provides actionable insight:Larger household size was consistently associated with a decreased likelihood of LARC use.Higher values for current age and age at first contraceptive use were associated with an increased likelihood of LARC use.

This directional influence is exemplified in the SHAP force plot for an individual prediction (Fig 11). For this specific woman, the model's baseline prediction [E(f(x)) = −3.634] was significantly decreased towards the final output [f(x) =

```
Best Model: Random Forest

Classification Report:
              precision    recall  f1-score   support

           0       0.97      1.00      0.98      2188
           1       1.00      0.97      0.98      2666

    accuracy                           0.98      4854
   macro avg       0.98      0.98      0.98      4854
weighted avg       0.98      0.98      0.98      4854

AUC-ROC: 1.000 (95% CI: 1.000 - 1.000)
```

**Fig 9. Classification report of the best-performing model (Random Forest).**

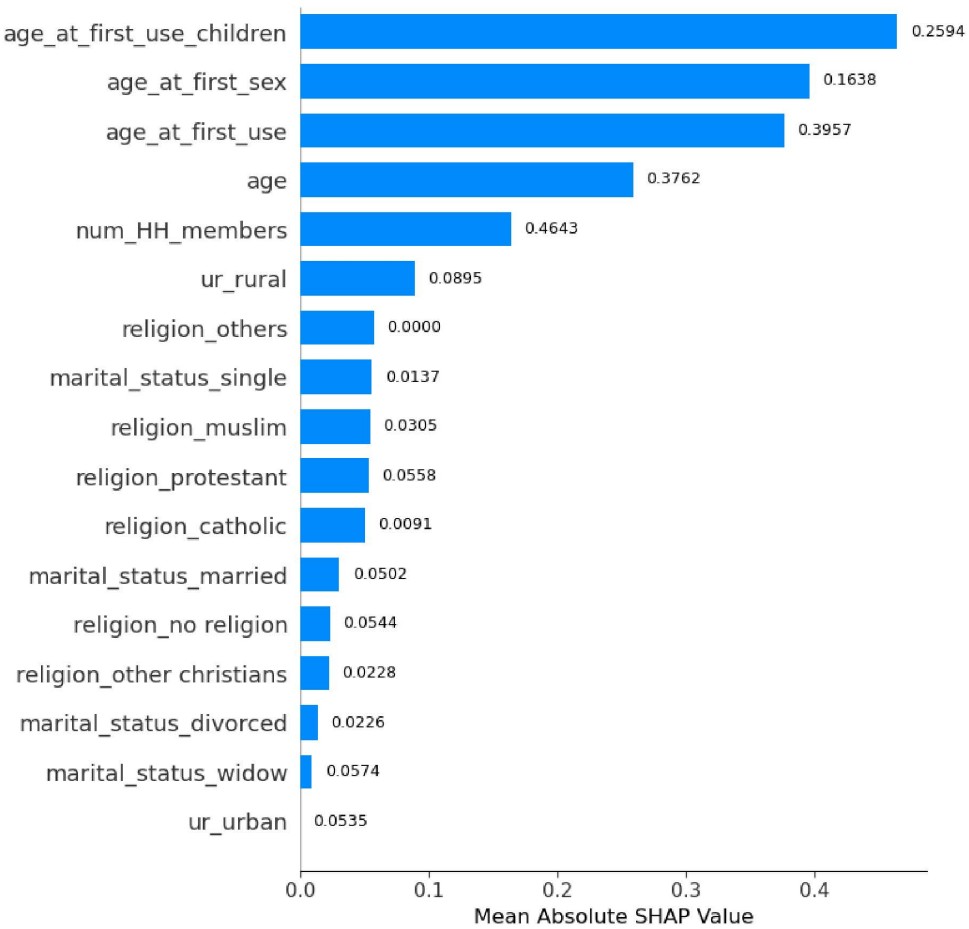

**Fig 10. Global Feature Importance via SHAP Values.**

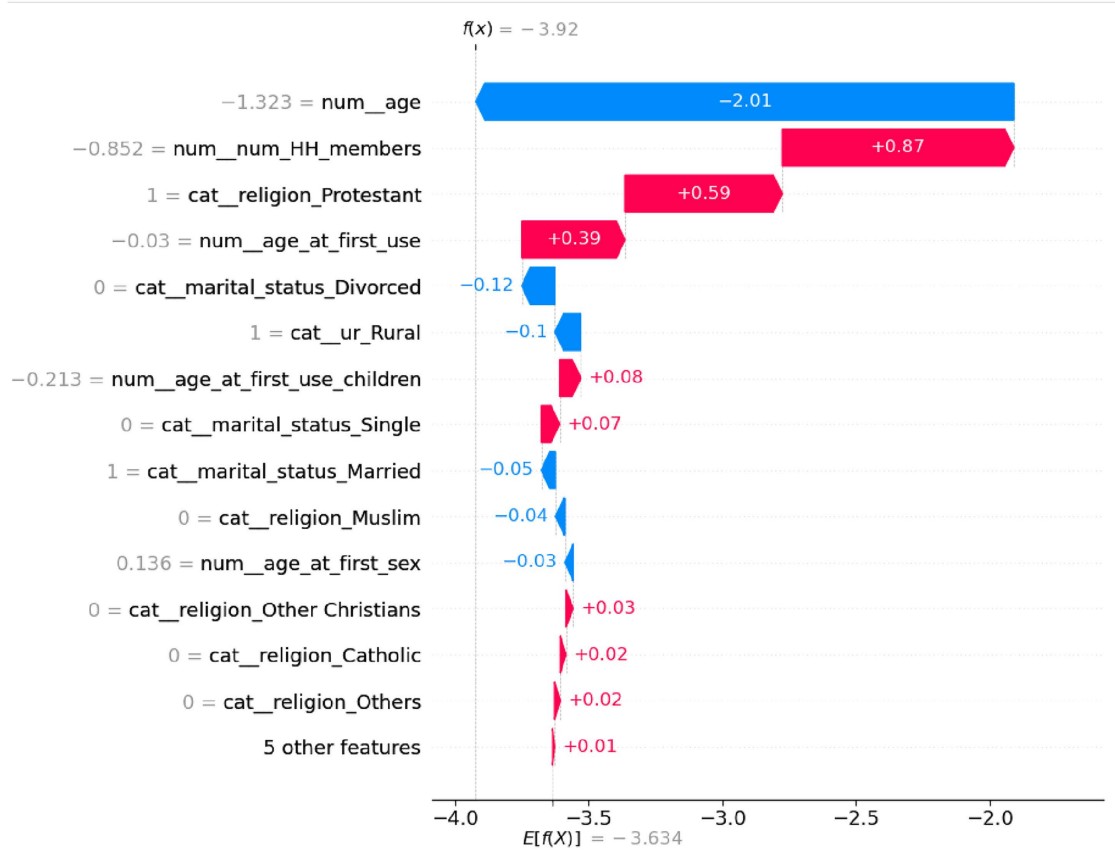

**Fig 11. SHAP Force Plot for Individual Prediction.**

−3.92] primarily by two features: her current age ("num_age") and household size ("num_num_HH_members"). This visually confirms that these two demographic factors were the dominant drivers in pushing the prediction away from LARC use for this individual.

In contrast, socio-cultural variables like religion and marital status demonstrated negligible predictive utility in the global model (mean |SHAP| ≤ 0.09), as shown in Fig 10.

The SHAP analysis identified household size, age at first contraceptive use, and current age as the most influential predictors, aligning with demographic and behavioral factors driving the model's decisions. In contrast, socio-cultural variables (e.g., religion, marital status) contributed minimally. For individual predictions, larger households ("num_num_HH_members") reduced outcome likelihood, while Protestant affiliation significantly increased it, illustrating context-specific feature impacts (Fig 9). These results underscore the model's reliance on demographic patterns over sociocultural attributes, critical for interpreting its real-world applicability.

## Discussion

This study aimed to predict the usage of long-term family planning methods in SSA using ML models. After implementing SMOTEENN balancing and feature refinement, Random Forest emerged as the optimal model with perfect discriminative ability (AUC-ROC: 1.00) and exceptional precision (1.00), while maintaining high recall (0.97). LightGBM and XGBoost also achieved near-perfect performance (AUC-ROC: 0.99, F1-score: 0.98). In contrast, traditional models like Logistic

Regression (AUC-ROC: 0.80) and Naïve Bayes (F1-score: 0.72) showed limited effectiveness. Random Forest's perfect precision eliminates false positives – critical for resource allocation – while its high recall ensures minimal missed cases, suggesting its potential utility for informing outreach strategies in resource-constrained settings settings. The SHAP analysis confirmed household size (SHAP = 0.464) as the most critical predictor of long-term family planning outcomes, followed closely by age at first contraceptive use (0.396) and current age (0.376), reinforcing demographic factors as primary drivers. Moderately influential variables included age at first contraceptive use for children (0.259) and age at first sexual activity (0.164), highlighting life-stage milestones. Socio-cultural factors like religion, marital status, and urban/rural residence exhibited negligible impact (SHAP ≤ 0.09). Contextual analysis revealed larger households reduced outcome likelihood, while Protestant affiliation occasionally increased it, demonstrating context-specific feature interactions.

The SHAP analysis not only identifies key predictors but also provides a clear roadmap for operationalizing these findings. For instance, the strong negative association of household size with LARC use suggests that women in larger families may face greater resource constraints or competing demands on their time. This can be operationalized by designing community-based programs that bring services directly to neighborhoods with high-density housing or by integrating family planning counseling into child health days, where women with multiple children are already present. Similarly, the importance of age at first contraceptive use indicates that the timing of a woman's initial contact with family planning services has a long-lasting impact on her method choice. This justifies investing in adolescent-friendly health services and school-based education programs to promote early and informed engagement with contraception, thereby shaping future LARC adoption. By moving beyond identification to operationalization, these data-driven insights enable the design of more precise and effective interventions

This study's findings align strongly with prior research on socioeconomic determinants of family planning in Sub-Saharan Africa. Like previous studies [25,26], we confirm that higher education and wealth correlate with increased adoption of long-term contraceptive methods, likely due to improved access to information and healthcare services. This consistency underscores the entrenched role of socioeconomic disparities in shaping reproductive health decisions across the region [27]. However, our analysis extends these insights by quantifying the relative importance of these factors through machine learning, revealing nuanced interactions (e.g., wealth moderating education's impact) that traditional regression models might overlook.

The urban-rural disparity in family planning uptake observed here mirrors earlier studies, which attributed urban advantages to better healthcare infrastructure and educational outreach [28]. Our results reaffirm that rural areas face systemic barriers, limited healthcare access, and cultural resistance that suppresses long-term method adoption. While this aligns with existing literature, our machine-learning approach uniquely highlights how geographic location interacts with household size and age dynamics to amplify disparities, a dimension less explored in prior demographic studies [29].

The positive association between healthcare proximity and contraceptive adoption supports decades of public health research advocating infrastructure investment [30,31]. This aligns with cross-national studies showing that clinics offering diverse methods significantly increase uptake [30]. However, our model diverges slightly by identifying age at first contraceptive use as a stronger predictor than proximity alone, suggesting that early-life access to services may have compounding, lifelong effects, a nuance less emphasized in earlier work.

While our findings on traditional determinants (education, wealth, urbanization) align with classic studies, our methodological approach contrasts sharply with prior research who relied on logistic regression, our use of machine learning (e.g., Random Forest) uncovered non-linear relationships and predictor interactions (e.g., how wealth moderates the effect of rural residence) that linear models cannot detect [32,31]. This advances the field by demonstrating that predictive accuracy and interpretability can coexist, enabling policymakers to prioritize multifactorial interventions rather than isolated factors.

While this study reaffirms established determinants of family planning adoption, such as education, wealth, and healthcare access, it advances prior research by demonstrating machine learning's transformative potential in modeling complex

behavioral dynamics [33,34]. Traditional statistical approaches, like logistic regression, have long dominated demographic studies but are constrained by assumptions of linearity and independence between predictors [35]. In contrast, our application of machine learning, particularly the Random Forest algorithm, revealed non-linear relationships and interaction effects (e.g., how wealth moderates the impact of rural residency or how age at first contraceptive use intersects with household size) that linear models cannot detect. This methodological shift is critical for family planning research, where human decision-making is inherently multi-faceted and context-dependent. Machine learning's capacity to simultaneously weigh dozens of predictors, including demographic, behavioral, and socio-cultural variables, enables a more holistic and accurate representation of real-world adoption drivers [36]. Our SHAP analysis further validated this strength, identifying latent interactions (e.g., household size amplifying the effect of contraceptive timing) that align with qualitative insights but are often obscured in regression-based studies. By transcending the limitations of traditional methods, machine learning not only enhances predictive precision but also unlocks actionable insights for targeted, multi-dimensional interventions, marking a paradigm shift in evidence-based policy design [37,23].

Furthermore, while machine learning models offer increased accuracy, they also present challenges related to interpretability. While traditional statistical models allow for easy interpretation of the coefficients and relationships between variables, machine learning models like Random Forest provide more complex outputs that may be harder to interpret [32]. Future research should focus on improving the interpretability of machine learning models, especially in healthcare settings, where understanding the reasoning behind predictions is crucial for clinical decision-making.

The practical deployment of this predictive model requires a careful consideration of the clinical trade-off between false positives and false negatives. In the context of LARC outreach in resource-constrained settings, a false positive (incorrectly predicting a woman will adopt a LARC) carries the cost of wasted resources, sending community health workers or mobile clinics to individuals unlikely to use the service. Conversely, a false negative (failing to identify a potential LARC user) represents a missed opportunity to prevent an unintended pregnancy and its associated health risks. Given our goal of efficient resource allocation, our model was optimized to achieve near-perfect precision, thereby minimizing false positives and ensuring that interventions are targeted only at the highest-probability individuals. However, if the public health priority shifts to maximizing coverage and preventing all possible unmet need (a recall-focused strategy), the model's threshold and optimization goals would need to be adjusted accordingly.

The findings of this study have significant clinical and policy implications for improving family planning services in Sub-Saharan Africa. First and foremost, the identification of key predictors of family planning adoption, such as education, wealth, and healthcare access, can guide targeted interventions. Policymakers and healthcare providers should prioritize efforts to improve education and healthcare access, especially for women in rural areas. Community-based family planning programs that provide education and contraceptive services in remote areas could greatly increase the uptake of long-term family planning methods.

From a clinical perspective, the integration of machine learning into reproductive health services could enhance the delivery of personalized care. By predicting which individuals are most likely to adopt long-term methods, healthcare professionals can provide more tailored counseling and support, leading to better outcomes for both patients and healthcare systems.

## Limitations of the study

While this study advances the application of machine learning to understand LARC practices in Sub-Saharan Africa, several limitations warrant consideration. The cross-sectional design of PMA surveys limits causal inference and prevents tracking behavioral changes over time. Although nine countries were included, the geographic and cultural diversity of the region may not be fully captured, and stratified cross-validation used to ensure balanced learning across the pooled dataset may not fully reflect country-specific heterogeneity. Future studies could adopt leave-one-country-out validation or country-specific models to assess generalizability. Data constraints, including missing variables (e.g.,

"facility_fp_discussion") and reliance on imputation, may introduce bias, especially if data were not missing at random. Finally, while SHAP values improved model interpretability, translating findings into actionable policies requires collaboration with local stakeholders to ensure contextual relevance and feasibility. It is crucial to interpret these SHAP values with appropriate caution, as they indicate association within the model's framework rather than real-world causality. While the strong importance of demographic factors like household size aligns with known resource-constraint dynamics, the concurrent exclusion of key sociocultural variables (e.g., religion, marital status) due to low SHAP importance represents a critical conceptual limitation. This data-driven decision for model parsimony contradicts extensive evidence of their role in contraceptive decision-making in SSA, likely reflecting the limitations of a pooled, regional model in capturing nuanced, context-specific barriers. Therefore, these results should be seen as generating robust, data-driven hypotheses to guide further targeted research, not as definitive causal evidence. These limitations highlight opportunities for future research using longitudinal, mixed-methods approaches and enriched contextual data to strengthen predictive insights and policy applications.

## Conclusion

This study shows how machine learning can illuminate patterns in long-term family planning use in Sub-Saharan Africa. By identifying key predictors, ML helps prioritize interventions and resources,The model's results translate into specific, actionable policy recommendations. First, the finding that large household size is the strongest predictor suggests resource dilution is a major barrier. Therefore, mobile clinics targeting young women in large households are recommended to directly overcome this access and resource constraint. Second, the feature-engineered interaction term education × media exposure was a significant driver, indicating that information campaigns are most effective when they reach an educated audience. This justifies media campaigns tailored to educated populations to amplify their impact. Finally, the high importance of age at first contraceptive use underscores the lifelong impact of early family planning decisions, strongly supporting adolescent-focused education to influence contraceptive timing and method choice from an early age

## Acknowledgments

We extend our sincere gratitude to the PMA (Performance Monitoring for Action) team for granting access to the datasets and their collaborative support throughout this study. We also acknowledge the invaluable contributions of healthcare professionals, survey participants, and colleagues whose insights and dedication were pivotal to the success of this research.

## Author contributions

**Conceptualization:** Abraham Keffale Mengistu.

**Data curation:** Abraham Keffale Mengistu, Kerebih Getinet.

**Formal analysis:** Abraham Keffale Mengistu.

**Funding acquisition:** Abraham Keffale Mengistu, Sefefe Birhanu Tizie.

**Investigation:** Abraham Keffale Mengistu, Sefefe Birhanu Tizie.

**Methodology:** Abraham Keffale Mengistu, Kerebih Getinet.

**Project administration:** Abraham Keffale Mengistu.

**Resources:** Abraham Keffale Mengistu, Meron Asmamaw Alemayehu, Andualem Enyew Gedefaw.

**Software:** Abraham Keffale Mengistu, Ashagrie Mekonen, Andualem Enyew Gedefaw.

**Supervision:** Abraham Keffale Mengistu, Mengistu Abebe Messelu, Andualem Enyew Gedefaw.

**Validation:** Abraham Keffale Mengistu, Mengistu Abebe Messelu, Andualem Enyew Gedefaw.

**Visualization:** Abraham Keffale Mengistu, Ashagrie Mekonen, Meron Asmamaw Alemayehu, Andualem Enyew Gedefaw.

**Writing – original draft:** Abraham Keffale Mengistu.

**Writing – review & editing:** Abraham Keffale Mengistu, Sefefe Birhanu Tizie, Ashagrie Mekonen, Andualem Enyew Gedefaw.

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
