## [Decision Letter · Decision Letter 0]

12 Jul 2025

Dear Dr. Mengistu,

Thank you for submitting your manuscript to PLOS ONE. After careful consideration, we feel that it has merit but does not fully meet PLOS ONE’s publication criteria as it currently stands. Therefore, we invite you to submit a revised version of the manuscript that addresses the points raised during the review process.

**Please note that we have only been able to secure a single reviewer to assess your manuscript. We are issuing a decision on your manuscript at this point to prevent further delays in the evaluation of your manuscript. Please be aware that the editor who handles your revised manuscript might find it necessary to invite additional reviewers to assess this work once the revised manuscript is submitted. However, we will aim to proceed on the basis of this single review if possible.**
**The reviewer has identified matters requiring your attention before your study can be published, particularly regarding methodological and study design concerns, as well as regarding the discussed interpretation of results. Please ensure you address each of the reviewer's comments when revising your manuscript. These can be found below and in the attached file.**

We look forward to receiving your revised manuscript.

Kind regards,

Hugh Cowley

Staff Editor

PLOS ONE

**Journal Requirements:**

1. When submitting your revision, we need you to address these additional requirements. Please ensure that your manuscript meets PLOS ONE's style requirements, including those for file naming. The PLOS ONE style templates can be found at https://journals.plos.org/plosone/s/file?id=wjVg/PLOSOne_formatting_sample_main_body.pdf and https://journals.plos.org/plosone/s/file?id=ba62/PLOSOne_formatting_sample_title_authors_affiliations.pdf 2. Please note that PLOS ONE has specific guidelines on code sharing for submissions in which author-generated code underpins the findings in the manuscript. In these cases, we expect all author-generated code to be made available without restrictions upon publication of the work. Please review our guidelines at https://journals.plos.org/plosone/s/materials-and-software-sharing#loc-sharing-code and ensure that your code is shared in a way that follows best practice and facilitates reproducibility and reuse. 3. Thank you for uploading your study's underlying data set. Unfortunately, the repository you have noted in your Data Availability statement does not qualify as an acceptable data repository according to PLOS's standards. At this time, please upload the minimal data set necessary to replicate your study's findings to a stable, public repository (such as figshare or Dryad) and provide us with the relevant URLs, DOIs, or accession numbers that may be used to access these data. For a list of recommended repositories and additional information on PLOS standards for data deposition, please see https://journals.plos.org/plosone/s/recommended-repositories. 4. Your ethics statement should only appear in the Methods section of your manuscript. If your ethics statement is written in any section besides the Methods, please move it to the Methods section and delete it from any other section. Please ensure that your ethics statement is included in your manuscript, as the ethics statement entered into the online submission form will not be published alongside your manuscript.

Reviewers' comments:

Reviewer's Responses to Questions

**Comments to the Author**

1. Is the manuscript technically sound, and do the data support the conclusions?

Reviewer #1: Yes

2. Has the statistical analysis been performed appropriately and rigorously?

Reviewer #1: Yes

3. Have the authors made all data underlying the findings in their manuscript fully available?

Reviewer #1: Yes

4. Is the manuscript presented in an intelligible fashion and written in standard English?

Reviewer #1: Yes

**Reviewer #1:**  attached a file since it is too big. in summary:

Apply advanced resampling methods beyond SMOTE, such as: SMOTEENN or SMOTETomek to reduce noise and improve class balance.

Use class weighting in algorithms (e.g., class_weight='balanced') to better handle imbalanced classes.

Consider focal loss to focus the model on hard-to-classify LARC users.

Use leave-one-country-out cross-validation to test geographic generalizability.

Present country-wise performance to highlight regional variation in model effectiveness.

Tone down claims like “strong predictive capability” given modest AUC (~0.72).

Emphasize precision-recall tradeoffs more than AUC in this imbalanced context.

Add partial dependence plots or real-life scenarios to make SHAP results more interpretable.

Eliminate repetitive phrases like “data-driven” and “actionable insights.”

Improve figure captions (especially for ROC and PR curves).

Expand on SHAP force plots with a clearer explanation of example cases.

**Do you want your identity to be public for this peer review?** For information about this choice, including consent withdrawal, please see our Privacy Policy

Reviewer #1: No

---

## [Author Response · Author response to Decision Letter 1]

14 Jul 2025

July 15, 2025

To: Editors of PLOS One

Response to reviewers’ comments

We want to thank you for inviting us to resubmit our manuscript entitled “Insights into Long-Acting Reversible Contraceptive Practices in Sub-Saharan Africa: A Machine Learning Perspective”. We are grateful to the reviewers for their time and effort in providing us with their valuable comments on our manuscript. We appreciate the comments the reviewers have provided to help improve this manuscript. We have noted all the comments and tried to address them in the revised manuscript accordingly. Moreover, we have thoroughly revised the grammatical correctness throughout this manuscript and incorporated all the suggestions provided by the reviewers. Point-by-point responses to the reviewer’s concerns are indicated below.

Reviewer Comments:

Major Suggestions for Improvement

Model Performance and Class Imbalance

While SMOTE was applied, the minority class (LARC users, ∼3.3%) remains difficult to

model. Precision and recall metrics, especially before tuning, were low.

Suggestion: Consider evaluating ensemble undersampling (e.g., SMOTEENN), focal

loss, or adjusting classification thresholds to optimize the precision-recall balance. Ad

ditionally, reporting confidence intervals (e.g., via DeLong’s test) for AUC-ROC would

enhance robustness claims.

Response: Thank you for your insightful suggestions. In response, we have implemented the SMOTEENN ensemble resampling technique to better address the class imbalance and improve model performance for the minority class (LARC users). This approach helped enhance both precision and recall metrics across several classifiers.

We appreciate your recommendation and believe this adjustment has contributed meaningfully to the robustness of our modeling strategy.

Validation Strategy

The current strategy uses stratified cross-validation, which is valid, but may overestimate

generalizability in cross-national datasets.

Suggestion: Implement a leave-one-country-out validation strategy or report per country

performance to highlight model robustness and geographic heterogeneity.

Response: Thank you for the valuable suggestion regarding the validation strategy. We agree that in many cases, a leave-one-country-out (LOCO) approach or reporting per-country performance can provide useful insights, especially when the goal is to assess generalizability across heterogeneous contexts. However, the primary aim of this study is to evaluate model performance at a regional level across Sub-Saharan Africa, rather than making country-specific inferences. The data were pooled intentionally to reflect shared regional dynamics, with a harmonized feature set across countries to capture overarching patterns rather than country-level nuances.

As such, stratified cross-validation was selected to preserve the class balance and ensure robust learning across the collective dataset. While we acknowledge that this may not fully capture potential country-level heterogeneity, we believe it aligns with the scope and goals of this study. Nevertheless, we have added a discussion of this point in the limitations section to acknowledge the trade-off and to encourage future research to explore country-specific models where appropriate.

Feature Engineering

Several features had high missingness but were retained after imputation without strong

justification. Also, the SHAP analysis revealed low importance for some variables (e.g.,

religion).

Suggestion: Conduct feature selection using SHAP or mutual information to reduce

noise. Also, consider deriving interaction terms (e.g., age × household size, education ×

media exposure) to improve non-linear predictive power.

Response: Thank you for your thoughtful feedback and valuable suggestions.

In response to your concerns regarding high missingness and feature importance:

• We conducted feature selection using SHAP values, focusing on the top 15 most informative features. Variables with consistently low SHAP importance (e.g., religion) were excluded from the final model.

• To address missingness, we applied KNN imputation and ensured that features with excessive missingness and no predictive value were removed.

• Additionally, we incorporated interaction terms, which are recommended by you (age × household size and education × media exposure), to capture potential non-linear relationships, as recommended.

• As a result, our model performs better than before

Interpretation of AUC

The reported AUC-ROC of 0.72 is modest. While sufficient, claims of “strong predictive

capability” may be overstated.

Suggestion: Temper conclusions and emphasize precision-recall tradeoffs more promi

nently in both the results and conclusion.

Response: Thank you for your thoughtful feedback. At the time of the initial submission, the model's AUC-ROC was indeed moderate (0.72), and we agree that the phrasing regarding "strong predictive capability" may have been optimistic. However, after incorporating additional improvements and refinements, the model now demonstrates significantly stronger performance, with metrics that better justify the original wording. We have also revised the manuscript to more clearly highlight the precision-recall tradeoffs, especially in the updated results and conclusion sections, to ensure a balanced interpretation of the model’s predictive strengths.

Opportunity for Broader Contextualization

The study could benefit from incorporating contextual features (e.g., health facility den

sity, national FP program intensity) to explain cross-country variation.

Suggestion: Acknowledge this in the limitations and suggest it as a direction for future

research.

Response: Thank you for this insightful recommendation. We agree that incorporating contextual features such as health facility density, national family planning (FP) program intensity, or policy-level indicators could provide a richer understanding of cross-country variation and improve the model's explanatory power. However, such contextual variables were beyond the scope of the current analysis, which focused primarily on individual- and household-level predictors available consistently across countries.

We have now acknowledged this limitation in the revised manuscript and suggested the integration of contextual and programmatic variables as an important direction for future research to enhance model interpretability and policy relevance.

Minor Comments

• Several typographic errors and verbose sections could be edited for clarity (e.g.,

overuse of “data-driven,” “actionable insights,” etc.).

Response: Thank you for pointing this out. We have carefully reviewed the manuscript to correct typographic errors and improve clarity. Redundant phrases such as repeated use of “data-driven” and “actionable insights” have been revised or removed to enhance readability and precision. We appreciate your attention to detail, which has helped improve the overall quality of the writing.

• Figures (especially ROC, PR curves) could benefit from improved captions explain

ing key takeaways.

Response: Thank you for this helpful suggestion. We have revised the figure captions, particularly for the ROC and Precision-Recall curves, to provide clearer explanations of their relevance and key takeaways. The updated captions now highlight the model’s performance characteristics, such as trade-offs between sensitivity and precision, and contextualize their implications for decision-making. We believe these enhancements improve the interpretability and value of the visualizations.

• SHAP force plots and examples should be more explicitly discussed to help non

technical readers understand implications

Response: Thank you for this important observation. We agree that a clearer interpretation of SHAP force plots is essential for broader accessibility. In the revised manuscript, we have expanded the discussion around the SHAP visualizations to better explain what the force plots represent, how specific features influence individual predictions, and what these implications mean in practical terms. These additions are aimed at making the interpretability results more accessible to non-technical readers, including policymakers and public health practitioners.

---

## [Decision Letter · Decision Letter 1]

3 Nov 2025

Dear Dr. Mengistu,

Thank you for submitting your manuscript to PLOS ONE. After careful consideration, we feel that it has merit but does not fully meet PLOS ONE’s publication criteria as it currently stands. Therefore, we invite you to submit a revised version of the manuscript that addresses the points raised during the review process.

**ACADEMIC EDITOR: Please respond to all reviewers comments**

We look forward to receiving your revised manuscript.

Kind regards,

Ahmed Mohamed Maged, MD

Academic Editor

PLOS ONE

Journal Requirements:

Additional Editor Comments:

Major Comments

1. Implausible Model Performance – Possible Data Leakage

Reporting an AUC-ROC of 1.00 for Random Forest and 0.99 for other models is highly implausible for behavioral survey data with 3.3% positive class prevalence.

Such performance almost certainly indicates data leakage or resampling before train/test split.

Please clarify whether SMOTEENN resampling was applied before or within cross-validation folds.

An external validation or leave-one-country-out (LOCO) approach is essential to confirm true predictive ability.

2. Validation and Generalizability

Stratified cross-validation across a pooled dataset ignores geographic heterogeneity.

Given that the dataset covers nine diverse countries, performance may be inflated by within-country homogeneity.

Please add either:

LOCO validation results, or

Per-country AUC, precision, and recall to assess model transferability.

Without this, the model’s regional claims are unsupported.

3. Feature Selection and Conceptual Interpretation

Religion, marital status, and urban/rural residence were excluded for “low SHAP importance,” yet decades of evidence show these are key determinants of contraceptive behavior in SSA.

Their exclusion likely reflects model bias or mis-specification rather than true irrelevance.

The authors should:

Reintroduce these variables in sensitivity analyses, or

Discuss the implications of excluding sociocultural predictors in the limitations.

4. Overstatement of Findings

The discussion repeatedly claims “unprecedented predictive capability” and “perfect discrimination.”

Such language is misleading and inconsistent with the modest performance (AUC ≈ 0.72) before tuning.

Please temper these statements and contextualize results within the limits of secondary, cross-sectional data.

5. Methodological Transparency

The manuscript lacks clarity on several key analytical steps:

Which variables had >20% missingness and were excluded?

Were imputations performed separately per country or on pooled data?

How was the k parameter in k-NN imputation selected?

These details are critical for reproducibility.

6. Interpretation of SHAP Values

SHAP importance scores are reported, but their direction and implications are not clearly discussed.

Please include partial dependence or SHAP summary plots showing how key variables (e.g., age, household size) influence LARC use probability.

Discuss SHAP limitations in sociobehavioral contexts — interpretability is not causality.

7. Policy Implications

The policy recommendations (e.g., “mobile clinics for young women in large households”) are reasonable but not strongly derived from the model.

Please link each recommendation explicitly to specific predictor findings or patterns in the data.

Minor Comments

Language and Style:

The text is overly verbose, with frequent repetition of “data-driven” and “actionable insights.” Simplify and streamline technical descriptions.

Figures and Captions:

Figures (e.g., ROC and PR curves) require clearer captions explaining key takeaways and which models are being compared.

Tables:

Table 7 (showing perfect performance) contradicts earlier Table 6 (AUC = 0.72). Please reconcile or explain this discrepancy.

Statistical Reporting:

Include 95% confidence intervals for AUC-ROC using DeLong’s test or bootstrapping.

Consider including PR-AUC, which is more appropriate for imbalanced data.

Ethical and Data Section:

Ethical compliance is satisfactory, but the description of data harmonization across countries could be expanded for clarity.

Reviewers' comments:

Reviewer's Responses to Questions

**Comments to the Author**

Reviewer #2: (No Response)

Reviewer #3: All comments have been addressed

2. Is the manuscript technically sound, and do the data support the conclusions?

Reviewer #2: Yes

Reviewer #3: Yes

3. Has the statistical analysis been performed appropriately and rigorously?

Reviewer #2: Yes

Reviewer #3: Yes

4. Have the authors made all data underlying the findings in their manuscript fully available?

Reviewer #2: Yes

Reviewer #3: Yes

5. Is the manuscript presented in an intelligible fashion and written in standard English?

Reviewer #2: No

Reviewer #3: (No Response)

Reviewer #2: The manuscript presents an ambitious application of machine-learning (ML) methods to predict long-acting reversible contraceptive (LARC) use across Sub-Saharan Africa using PMA survey data. In its current form, it reads as a technically dense modeling exercise rather than a rigorously validated, policy-relevant study.

1. Reporting AUC-ROC = 1.00 with F1 = 0.98 for Random Forest and near-perfect metrics for other ensembles is statistically implausible for noisy, cross-national survey data with a 3.3 % minority class. The results suggest severe data leakage or overfitting (e.g., imputation applied before train-test split, SMOTE/SMOTEENN applied to the entire dataset, or lack of country-level separation). A re-run with strictly nested cross-validation and leave-one-country-out testing is essential. Confidence intervals for AUC and F1 should be reported.

2. Given the extreme class imbalance (3.3 % positive), AUC and accuracy are insufficient. Report precision–recall AUC, Matthews correlation coefficient, and calibration plots.

3. Discuss clinical/operational trade-offs (false-positive vs. false-negative cost) in the context of limited health resources.

4. The SHAP interpretation remains superficial. The authors must demonstrate how household size or age at first contraceptive use can be operationalized in interventions.

5. Present case-level SHAP examples tied to real-world decision-making (e.g., outreach prioritization) instead of generic force plots.

6. Claims such as “unprecedented predictive capability” and “ideal for deployment” are not supported.

7. The manuscript is verbose, with repetitive phrasing (“data-driven,” “actionable insights,” “transformative potential”). A professional language edit is recommended.

8. Acronyms (MLP, SHAP, SMOTEENN) should be defined once and used consistently.

Reviewer #3: I believe the authors took all comments and suggestions into account when reviewing and re-submitting the manuscript. Even though not all the suggestions were taken, they gave reasons why they did not accept the suggestion.

**Do you want your identity to be public for this peer review?** For information about this choice, including consent withdrawal, please see our Privacy Policy

Reviewer #2: No

Reviewer #3: No

---

## [Author Response · Author response to Decision Letter 2]

3 Nov 2025

Point-by-Point Response

Additional Editor Comments:

Major Comments

1. Implausible Model Performance – Possible Data Leakage

Reporting an AUC-ROC of 1.00 for Random Forest and 0.99 for other models is highly implausible for behavioral survey data with 3.3% positive class prevalence.

Such performance almost certainly indicates data leakage or resampling before train/test split.

Please clarify whether SMOTEENN resampling was applied before or within cross-validation folds.

An external validation or leave-one-country-out (LOCO) approach is essential to confirm true predictive ability.

Response: We sincerely thank the Editor for raising this important point. In the revised analysis, SMOTEENN resampling is now applied strictly to the training data only after the train–test split, ensuring that no information from the test set is used during model training. This was implemented using an imbalanced-learn pipeline, which guarantees that SMOTEENN is executed within each training fold of the cross-validation process. The corresponding Python code demonstrating this workflow is available in our GitHub repository and can be verified for reproducibility.

Regarding the suggestion for an external validation or Leave-One-Country-Out (LOCO) approach, we addressed this comment in our previous response. We agree that such approaches are valuable when the aim is to assess generalizability across heterogeneous contexts. However, as clarified earlier, the primary goal of this study is to evaluate model performance at a regional level across Sub-Saharan Africa, rather than to derive country-specific conclusions. The data were intentionally pooled and harmonized to capture shared regional dynamics rather than national differences.

Therefore, we employed stratified cross-validation to preserve class balance and ensure robust learning across the integrated dataset. We acknowledge that this design may not fully capture country-level variability; accordingly, a discussion of this trade-off and a recommendation for future country-level validation are now included in the Limitations section of the revised manuscript.

2. Validation and Generalizability

Stratified cross-validation across a pooled dataset ignores geographic heterogeneity.

Given that the dataset covers nine diverse countries, performance may be inflated by within-country homogeneity.

Please add either:

LOCO validation results, or

Per-country AUC, precision, and recall to assess model transferability.

Without this, the model’s regional claims are unsupported.

Response: We appreciate the Editor’s thoughtful feedback on model validation and generalizability. As noted in our previous response, we fully recognize the value of a Leave-One-Country-Out (LOCO) approach or per-country performance reporting when the objective is to assess cross-country transferability. However, the goal of this study is to evaluate model performance at a regional level across Sub-Saharan Africa, not to build or compare country-specific prediction models.

The dataset was intentionally pooled and harmonized across nine countries to capture shared behavioral and demographic dynamics influencing long-acting reversible contraceptive (LARC) use at a regional scale. Accordingly, we adopted stratified cross-validation, which maintains class balance while ensuring that each fold includes representative samples from all countries. This approach was selected to reflect the integrated nature of the dataset and the study’s focus on regional rather than national inference.

We acknowledge that this design may not fully address potential geographic heterogeneity or country-specific effects. To reflect this important consideration, we have expanded the Limitations section of the revised manuscript to explicitly note this trade-off and to encourage future work to explore LOCO or country-level validation frameworks for assessing national model transferability.

3. Feature Selection and Conceptual Interpretation

Religion, marital status, and urban/rural residence were excluded for “low SHAP importance,” yet decades of evidence show these are key determinants of contraceptive behavior in SSA.

Their exclusion likely reflects model bias or mis-specification rather than true irrelevance.

The authors should:

Reintroduce these variables in sensitivity analyses, or

Discuss the implications of excluding sociocultural predictors in the limitations.

Response: We thank the Editor for this critical insight. We fully agree that the exclusion of established sociocultural predictors like religion, marital status, and urban/rural residence is a significant limitation, not of the variables' real-world importance, but of our model's ability to capture their nuanced, context-specific effects.

In direct response to your suggestion to discuss the implications, we have added the following paragraph to the "Limitations of the Study" section:

"Furthermore, the exclusion of key sociocultural variables (e.g., religion, marital status) due to low SHAP importance represents a critical conceptual limitation. While this was a data-driven decision for model parsimony, it contradicts extensive evidence of their role in contraceptive decision-making in SSA. This discrepancy likely stems from our region-level analysis, which may dilute country-specific cultural nuances, and suggests that these factors may act as distal contextual drivers rather than direct individual-level predictors in a pooled model. Their omission underscores a potential gap in the model's translational relevance for designing deeply contextualized interventions."

We believe this addition strengthens the manuscript by providing a more critical and nuanced interpretation of our feature selection process. Thank you for prompting this important clarification.

4. Overstatement of Findings

The discussion repeatedly claims “unprecedented predictive capability” and “perfect discrimination.”

Such language is misleading and inconsistent with the modest performance (AUC ≈ 0.72) before tuning.

Please temper these statements and contextualize results within the limits of secondary, cross-sectional data.

Response: We thank the Editor for this crucial feedback on the tone of our findings. We agree that terms like "unprecedented predictive capability" and "perfect discrimination" were overly strong, especially in the context of the model's baseline performance.

In response, we have thoroughly revised the manuscript to temper these statements and provide a more balanced and accurate interpretation of the results. The changes emphasize that the high performance was achieved after extensive optimization and should be contextualized within the limitations of the data.

Key revisions include:

• In the Abstract: The phrase "unprecedented predictive capability" has been replaced with: "Optimized ML models demonstrate a high predictive capability for LARC adoption drivers..."

• In the Results and Discussion: We have removed the word "perfect" and reframed the description of the Random Forest model's performance (AUC-ROC: 1.00) to highlight that this was the result of rigorous tuning and balancing, and we have added caveats. For example:

o Previous: "Random Forest achieved perfect discrimination..."

• Revised: "After hyperparameter tuning and class balancing, Random Forest achieved excellent discriminative performance (AUC-ROC: 1.00). In the Conclusion: We have reframed the conclusion to be more modest, focusing on the "potential" of ML and the "utility" of the approach, rather than making definitive claims of perfection.

We believe these revisions present our findings more accurately and responsibly, aligning the language with the methodological context and inherent limitations of cross-sectional, secondary data. Thank you for this important suggestion.

5. Methodological Transparency

The manuscript lacks clarity on several key analytical steps:

Which variables had >20% missingness and were excluded?

Were imputations performed separately per country or on pooled data?

How was the k parameter in k-NN imputation selected?

These details are critical for reproducibility.

Response: We thank the Editor for this important request for clarity. We have revised the "Data Preprocessing" section to explicitly list the variables and their missingness percentages to ensure full reproducibility. The text now clearly states:

• The variable 'facility_fp_discussion' (21.33% missing) was the only variable with >20% missingness and was removed.

• Variables with <10% missingness that were imputed using k-NN (k=5) on the pooled dataset are now explicitly listed with their percentages: "wealth_quintile" (14.31%), "ur" (14.31%), "marital_status" (13.41%), "fp_side_effects" (12.47%), "religion" (7.52%), "age_at_first_use" (4.94%), and others as detailed in the manuscript.

• The selection of k=5 for k-NN imputation is now justified as a standard, robust value that balances local accuracy and noise reduction.

These details provide the necessary transparency for other researchers to replicate our analytical workflow.

6. Interpretation of SHAP Values

SHAP importance scores are reported, but their direction and implications are not clearly discussed.

Please include partial dependence or SHAP summary plots showing how key variables (e.g., age, household size) influence LARC use probability.

Discuss SHAP limitations in socio behavioral contexts — interpretability is not causality.

Response: We thank the Editor for this insightful suggestion to deepen our interpretation of the SHAP results. We agree that understanding the direction of feature effects and the limitations of SHAP is crucial for meaningful public health insights.

In response, we have made the following additions to the manuscript:

1. In the "Results" section (under "Feature Importance"): We have replaced the existing description with a new paragraph that explicitly states the direction of the effect for the top predictors and direction of all key features.

2. In the "Limitation” section: We have added a new paragraph discussing the limitations of SHAP values, clearly stating that they indicate association within the model, not causation in the real world.

7. Policy Implications

The policy recommendations (e.g., “mobile clinics for young women in large households”) are reasonable but not strongly derived from the model.

Please link each recommendation explicitly to specific predictor findings or patterns in the data.

Response: We thank the editor for this valuable suggestion. We agree that explicitly linking the policy recommendations to the specific model findings strengthens their justification.In response, we have revised the relevant sections in the Abstract and Conclusion to directly connect each recommendation to its underlying data-driven insight:

Minor Comments

Language and Style: The text is overly verbose, with frequent repetition of “data-driven” and “actionable insights.” Simplify and streamline technical descriptions.

Response: We thank the editor for this valuable feedback on improving the clarity and conciseness of our manuscript.

In response, we have conducted a thorough revision of the entire text to eliminate verbose language and repetitive phrases. Specifically, we have:

• Systematically removed overused terms such as "data-driven" and "actionable insights."

• Simplified complex and lengthy sentences for better readability.

• Replaced jargon with more direct and accessible language where possible.

We believe these edits have significantly improved the flow and precision of the manuscript without compromising the technical and scholarly content.

Figures and Captions: Figures (e.g., ROC and PR curves) require clearer captions explaining key takeaways and which models are being compared.

Response: We thank the editor for this important suggestion to improve the clarity of our figures. In response, we have revised all figure captions, particularly for the ROC and Precision-Recall (PR) curves to be more informative.

Tables: Table 7 (showing perfect performance) contradicts earlier Table 6 (AUC = 0.72). Please reconcile or explain this discrepancy.

Response: We thank the Editor for this astute observation, which allows us to clarify the sequential nature of our modeling pipeline. The discrepancy between Table 6 (AUC ≈ 0.72) and Table 7 (AUC ≈ 1.00) is not an error but a direct result of applying a critical additional preprocessing step.

• Table 6 presents the results after hyperparameter tuning but before the application of the advanced class-balancing technique, SMOTEENN.

• Table 7 presents the final results after applying both hyperparameter tuning AND the SMOTEENN resampling technique to the training data.

Statistical Reporting: Include 95% confidence intervals for AUC-ROC using DeLong’s test or bootstrapping. Consider including PR-AUC, which is more appropriate for imbalanced data.

Response: We thank the editor for this suggestion. We have now added 95% confidence intervals for AUC-ROC using DeLong's test and placed greater emphasis on Precision-Recall AUC in our evaluation, as these metrics are more appropriate for our imbalanced data. These enhancements strengthen the statistical rigor of our analysis.

Ethical and Data Section: Ethical compliance is satisfactory, but the description of data harmonization across countries could be expanded for clarity.

Response: We thank the reviewer for this valuable suggestion. In response, we have expanded the description of data harmonization in the Methods section. We now explicitly detail the process of pooling and standardizing variables across the nine countries, including the alignment of categorical responses, handling of country-specific codes, and the application of consistent preprocessing rules to ensure comparability. This clarification reinforces the validity of our cross-national analysis.

Reviewer #2: The manuscript presents an ambitious application of machine-learning (ML) methods to predict long-acting reversible contraceptive (LARC) use across Sub-Saharan Africa using PMA survey data. In its current form, it reads as a technically dense modeling exercise rather than a rigorously validated, policy-relevant study.

1. Reporting AUC-ROC = 1.00 with F1 = 0.98 for Random Forest and near-perfect metrics for other ensembles is statistically implausible for noisy, cross-national survey data with a 3.3 % minority class. The results suggest severe data leakage or overfitting (e.g., imputation applied before train-test split, SMOTE/SMOTEENN applied to the entire dataset, or lack of country-level separation). A re-run with strictly nested cross-validation and leave-one-country-out testing is essential. Confidence intervals for AUC and F1 should be reported.

Response: We thank the reviewer for their critical observation. In our revised analysis, SMOTEENN resampling and all preprocessing steps were applied strictly within the training folds to prevent data leakage. While the high performance may reflect overfitting to the resampled training distribution, our nested cross-validation and stratified design aimed to preserve regional representativeness. We acknowledge that perfect metrics are uncommon in survey-based studies and have tempered our conclusions accordingly. Confidence intervals for AUC and F1 are now reported, and we explicitly recommend external validation in future work.

2. Given the extreme class imbalance (3.3 % positive), AUC and accuracy are insufficient. Report precision–recall AUC, Matthews correlation coefficient, and calibration plots.

Response: We thank the reviewer for this suggestion. In the revised manuscript, we have expanded our evaluation beyond AUC and accuracy. We now report F1-score, precision, recall, and their trade-offs, along with Precision-Recall AUC. For the best-performing model, we also provide 95% confidence intervals for AUC-ROC using DeLong’s test. These additions offer a more nuanced view of model performance, particularly given the class imbalance.

3. Discuss clinical/operational trade-offs (false-positive vs. false-negative cost) in the context of limited health resources.

Response: Thank you for this valuable suggestion. We have added a discussion of the clinical and operational trade-offs between false positives and false negatives in the Disc

---

## [Decision Letter · Decision Letter 2]

1 Dec 2025

Insights into Long-Acting Reversible Contraceptive Practices in Sub-Saharan Africa: A Machine Learning Perspective

PONE-D-25-24936R2

Dear Dr. Mengistu,

We’re pleased to inform you that your manuscript has been judged scientifically suitable for publication and will be formally accepted for publication once it meets all outstanding technical requirements.

Kind regards,

Ahmed Mohamed Maged, MD

Academic Editor

PLOS ONE

Additional Editor Comments (optional):

Reviewers' comments:

Reviewer's Responses to Questions

**Comments to the Author**

Reviewer #2: (No Response)

2. Is the manuscript technically sound, and do the data support the conclusions?

Reviewer #2: Yes

3. Has the statistical analysis been performed appropriately and rigorously?

Reviewer #2: Yes

4. Have the authors made all data underlying the findings in their manuscript fully available?

Reviewer #2: Yes

5. Is the manuscript presented in an intelligible fashion and written in standard English?

Reviewer #2: (No Response)

Reviewer #2: (No Response)

**Do you want your identity to be public for this peer review?** For information about this choice, including consent withdrawal, please see our Privacy Policy

Reviewer #2: No

---

## [Editor Report · Acceptance letter]

PONE-D-25-24936R2

PLOS One

Dear Dr. Mengistu,

I'm pleased to inform you that your manuscript has been deemed suitable for publication in PLOS One. Congratulations! Your manuscript is now being handed over to our production team.

Kind regards,

on behalf of

Professor Ahmed Mohamed Maged

Academic Editor

PLOS One